# FAM122A ensures cell cycle interphase progression and checkpoint control by inhibiting B55α/PP2A through helical motifs

Jason S. Wasserman [1], Bulat Faezov[2,3], Kishan R. Patel[1], Alison M. Kurimchak [2], Seren M. Palacio [1], David J. Glass[2], Holly Fowle[1], Brennan C. McEwan [4], Qifang Xu [2], Ziran Zhao[1], Lauren Cressey [4], Neil Johnson [2], James S. Duncan[2], Arminja N. Kettenbach [4], Roland L. Dunbrack Jr [2] & Xavier Graña [1,2] ✉

The Ser/Thr protein phosphatase 2 A (PP2A) regulates the dephosphorylation of many phosphoproteins. Substrate recognition are mediated by B regulatory subunits. Here, we report the identification of a substrate conserved motif [RK]-V-x-x-[VI]-R in FAM122A, an inhibitor of B55α/PP2A. This motif is necessary for FAM122A binding to B55α, and computational structure prediction suggests the motif, which is helical, blocks substrate docking to the same site. In this model, FAM122A also spatially constrains substrate access by occluding the catalytic subunit. Consistently, FAM122A functions as a competitive inhibitor as it prevents substrate binding and dephosphorylation of CDK substrates by B55α/PP2A in cell lysates. FAM122A deficiency in human cell lines reduces the proliferation rate, cell cycle progression, and hinders G1/S and intra-S phase cell cycle checkpoints. FAM122A-KO in HEK293 cells attenuates CHK1 and CHK2 activation in response to replication stress. Overall, these data strongly suggest that FAM122A is a short helical motif (SHeM)-dependent, substrate-competitive inhibitor of B55α/PP2A that suppresses multiple functions of B55α in the DNA damage response and in timely progression through the cell cycle interphase.

Protein phosphorylation is a reversible post-translational modification that plays critical roles in the regulation of many signaling pathways and cellular processes. It is estimated that two-thirds of all human proteins are phosphorylated, with more than 98% of phosphorylation events occurring on serine and threonine residues[1–3]. Protein kinases catalyze the transfer of the γ-phosphate of ATP to phosphorylatable residues, while phosphatases hydrolyze the phosphate from phosphorylated residues. Members of the serine/threonine phosphoprotein phosphatase (PPP) family are responsible for the majority of dephosphorylation occurring in eukaryotic cells. Within this family,

protein phosphatase 1 (PP1) and protein phosphatase 2A (PP2A) account for more than 90% of the total phosphatase activity[4,5]. The PP2A Ser/Thr phosphatase forms multimeric complexes in cells. The heterodimeric "core enzyme" is made of a scaffold subunit (A) and a catalytic (C) subunit. For the formation of the heterotrimeric holoenzyme, the core dimer recruits a B subunit, which is the key determinant of substrate specificity for the holoenzyme reviewed in refs. 5–9. There are four B families: B/B55, B′/B56, B″/B72, and B‴/B93; each consisting of three to five isoforms and splice variants. The mechanism of substrate recognition by serine/threonine phosphatases is typically

[1]Fels Cancer Institute for Personalized Medicine. Temple University Lewis Katz School of Medicine, Philadelphia, PA, USA. [2]Fox Chase Cancer Center, Temple Health, Philadelphia, PA, USA. [3]Institute of Fundamental Medicine and Biology, Kazan Federal University, Kazan, Russian Federation. [4]Norris Cotton Cancer Center, Geisel School of Medicine at Dartmouth, Medical Center Drive, Lebanon, NH, USA. ✉e-mail: xgrana@temple.edu

dependent on short linear motifs (SLiMs) present in substrates, which enable a direct interaction with catalytic or regulatory subunits. For instance, the B56 family of B subunits bind substrates with a LxxIxE SLiM in Intrinsically Disordered Regions (IDRs)[10–12]. In contrast, PP2A/B55 substrate recruitment was thought to be mediated by charge-charge interactions between the surface of B55α and its substrates[13]. However, we have recently made an important discovery that challenges this view with significant implications for our understanding of PP2A biology, as PP2A/B55α is the most abundant PP2A holoenzyme. We identified a conserved SLiM [**RK**]-**V**-x-x-[**VI**]-**R** present in the intrinsically disordered regions (IDR) in a range of proteins, including substrates such as the retinoblastoma-related protein p107 and TAU[14].

B55α, which is the most abundant isoform of the B/B55 family, has been implicated in the dephosphorylation of many substrates with functions in cellular signaling, the cell cycle, and other cellular processes in post-mitotic cells[6]. PP2A/B55α holoenzymes are inactivated by a rapid switch at the G2/M transition that results in simultaneous activation of Cyclin B/CDK1. Inhibition of B55α is mediated by ARPP19 upon phosphorylation by a kinase called MASTL[15,16]. We have shown that PP2A/B55α is also key for an interphase equilibrium that modulates the phosphorylation state of the retinoblastoma-related proteins[17–19], and has also been implicated in checkpoint control[20,21]. Our search for B55α binding proteins that contain a degenerate B55α substrate SLiM identified FAM122A, a protein recently found to inhibit PP2A/B55α[22]. FAM122A has recently been shown to control the activation of PP2A/B55α in response to massive replication stress caused by CHK1 inhibition in NSCLC cell lines, which activates the WEE1 kinase, a negative regulator of CDK1[23]. However, the mechanism by which FAM122A inhibits the PP2A/B55α holoenzyme is not well understood.

Using rigorous biochemical and molecular modeling approaches, we have generated a strong body of data supporting substrate competition as a mechanism for inhibition of PP2A/B55α. In addition, we report that ablation of FAM122A in cells inhibits proliferation, attenuates mitogenic signaling linked to the G0/G1 transition, and abrogates the G1/S checkpoint in response to nucleotide depletion by a mechanism associated with attenuation of CHK1 and CHK2 signaling leading to DNA damage and cell death. Therefore, FAM122A controls PP2A/B55α function using a decoy mechanism to block substrate access to the enzyme active site thereby controlling cell cycle transitions stimulated by mitogens and genotoxic stresses.

## Results

### A filtered proteome-wide search identifies FAM122A as a potential SLiM-containing PP2A/B55α interactor

Using extensive mutational analysis and competition assays, we have previously defined the amino acid residues within a SLiM required for substrate binding to B55α. In the p107 model substrate, these residues direct the dephosphorylation of a proximal phosphosite. We have also shown that the SLiM is conserved in the unrelated substrate TAU and is required for its dephosphorylation. This allowed us to define an initial consensus B55 substrate SLiM, 'p[ST]-P-x(4,10)-[RK]-V-x-x-[VI]-R'[14]. In this report, we have used a proteome-wide search tool, ScanProsite[24,25], to identify potential novel PP2A/B55 substrates with a degenerate version of this SLiM (p[ST]-P-x(4,10)-[RK]-[VIL]-x-x-[VIL]-[RK]) that includes potential conservative amino acid variants (Fig. 1A, Supplementary Data 1). This search yielded 275 proteins (1.3% of the proteome). We then filtered the list of potential substrate candidates by determining which of these proteins have been detected by B55α pulldown proteomics[10], identified in phosphoproteomic analyses where B55α activity was inhibited in lysates[1], or in phosphoproteomic datasets of proteins dephosphorylated following doxycycline-inducible expression of FLAG-B55α in HEK293 cells (Supplementary Data 2).

A recurring 'hit' among the datasets was FAM122A (Fig. 1B, Supplementary Data 1), also known as PABIR1 or PPP2R1A-PPP2R2A-interacting phosphatase regulator 1, a highly conserved protein of

predicted disorder and a proposed inhibitor of the PP2A/B55α holoenzyme[22]. FAM122A exhibits a potential SLiM sequence, **RL**HQ**IK**, was identified in at least two independent B55α pulldown datasets[10,17], and exhibits several p-SP sites that are downregulated following upregulation of B55α in HEK293 and PC3 cells (Fig. 1C), including a proximal phospho-SP amino terminal from the potential SLiM sequence. Of note, a B55α pulldown dataset in Rat Chondrosarcoma cells, which we had previously reported, showed that FAM122A is the most abundant protein in B55α complexes other than the holoenzyme subunits[17]. Ontology analysis revealed that *FAM122A* originated by duplication of an ancestral *FAM122B* detected in *bilateria* present in worms, flies, frogs, and fish (Supplementary Fig. 1A). Amino acid sequence conservation shows that the SLiM sequence identified in FAM122A is conserved even in distant worms and flies (Fig. 1D; Supplementary Fig. 1B), which also express B55/PP2A holoenzyme subunits and in the two FAM122A paralogs, FAM122B and FAM122C. Altogether, these data suggest that FAM122A is a potential SLiM-dependent inhibitor.

### FAM122A is a SLiM-dependent interaction partner of B55α

Since FAM122A has been reported to be a bona fide inhibitor of PP2A/B55α[22], we determined if its interaction with B55α is dependent on functional SLiM sequences via immunoprecipitation of Myc-B55α and FLAG-FAM122A tagged mutants transfected in HEK293T cells. Mutation of murine mFLAG-FAM122A R81/L82 or I85/K86 SLiM residues to Ala abolished binding to Myc-B55α in reciprocal immunoprecipitations (Fig. 2A). In contrast, an S73A mutation, abolishing a phosphosite amino-terminal to the SLiM, had no effect (Fig. 2A, Supplementary Fig. 2A). Consistent with a functional SLiM, the B55α D197K mutant, which does not bind B55α substrates[14], also failed to bind FLAG-FAM122A (Fig. 2A). D197 is located in a deep groove on the surface of B55α, adjacent to the active site of the catalytic subunit. To rule out the possibility of the positively charged residues solely being responsible for the interaction, non-polar residues, Leu and Ile, were individually mutated to Ala in a GST-FAM122A deletion construct and subject to a GST-pulldown assay using HEK293T lysates. As shown in Fig. 2B, both L82 and I85 are critical for binding the holoenzyme in vitro. However, a short FAM122A$_{L70-K86}$ peptide fused to GST containing the '**RL**HQ**IK**' SLiM was not sufficient to bind the holoenzyme in vitro (Supplementary Fig. 2B), suggesting that additional residues contribute to binding. Thus, we next determined if other conserved regions in FAM122 family members (FR) were required or contributed to binding. FRs were selected by aligning the human FAM122A paralogs, FAM122B and FAM122C and denoting regions of conservation (Fig. 2C). Figure 2D schematically details the GST constructs used in pull-down assays of HEK293T lysates. Conserved region FR4, which contains the SLiM, is sufficient for binding, while regions FR1, FR2, and FR3, fail to bind B55α, but FR2 and FR3 significantly increase the binding avidity of FR4 (Fig. 2D–F). As charge-charge dynamic interactions in PP2A/B56 substrates increase binding avidity to B56[26], we mutated conserved positive residues in vertebrate FAM122A. Mutation of these conserved Arginine residues in FR2, FR3, and FR4 reduced binding to B55α (Supplementary Fig. 2C). Taken together, the interface of FAM122A:B55α binding is mediated via the SLiM and enhanced, at least in part by additional interactions of charged residues in FAM122A.

### AlphaFold2 predicts that the hFAM122A SLiM folds as a short helix, followed by a longer C-terminal helix, and that R84 (corresponding to murine R81), contacts B55α D197

Despite previous computational metrics denoting disorder in FAM122A, AlphaFold2 (AF2), an artificial intelligence (AI) program that performs predictions of protein structure[27], predicts a helix in FAM122A consisting of residues R84-G93 that encompasses the SLiM. This is followed by a C-terminal helix$_{L95-S120}$ (scoring confident to very high confident prediction, Fig. 3A). The rest of FAM122A was predicted

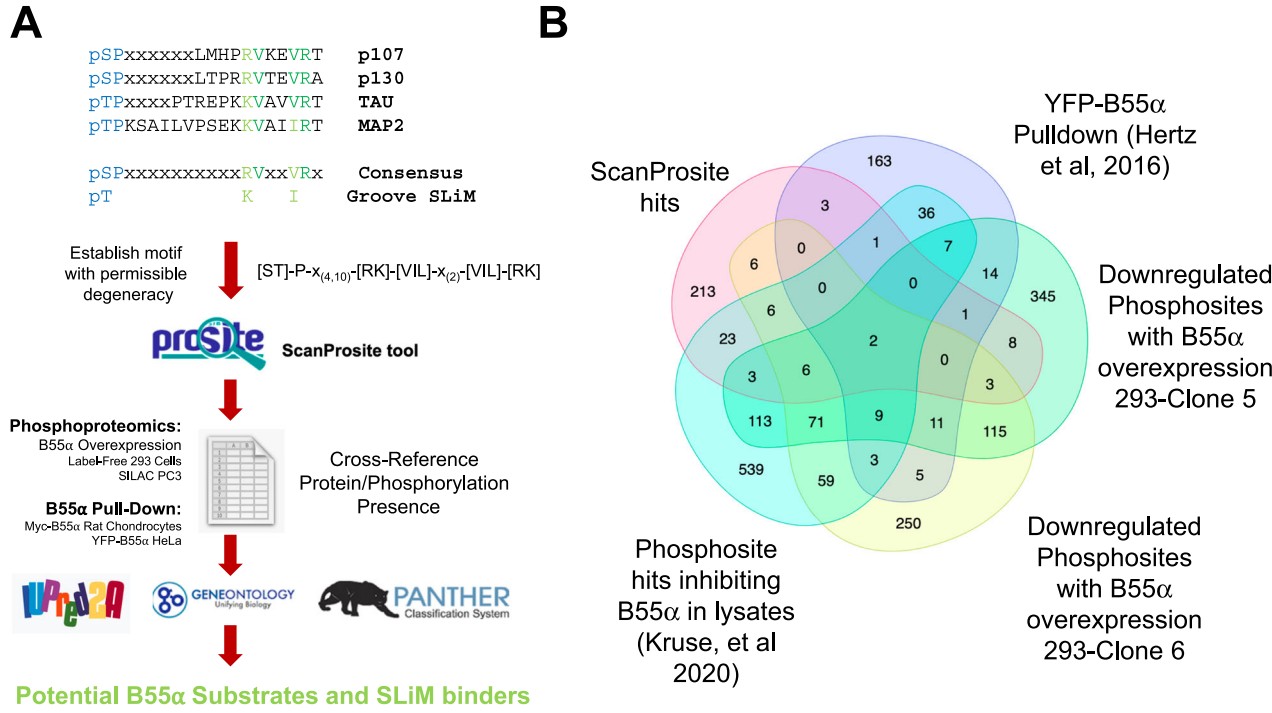

**A**

| | |
|---|---|
| pSPxxxxxxLMHPRVKEVRT | p107 |
| pSPxxxxxxLTPRRVTEVRA | p130 |
| pTPxxxxPTREPKKVAVVRT | TAU |
| pTPKSAILVPSEKKVAIIRT | MAP2 |
| | |
| pSPxxxxxxxxxRVxxVRx | Consensus |
| pT K I | Groove SLiM |

Establish motif with permissible degeneracy → [ST]-P-x$_{(4,10)}$-[RK]-[VIL]-x$_{(2)}$-[VIL]-[RK]

**ScanProsite tool**

**Phosphoproteomics:**
B55α Overexpression
Label-Free 293 Cells
SILAC PC3

**B55α Pull-Down:**
Myc-B55α Rat Chondrocytes
YFP-B55α HeLa

Cross-Reference Protein/Phosphorylation Presence

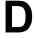 GENEONTOLOGY  PANTHER Classification System

**Potential B55α Substrates and SLiM binders**

**B**

ScanProsite hits

YFP-B55α Pulldown (Hertz et al, 2016)

Downregulated Phosphosites with B55α overexpression 293-Clone 5

Downregulated Phosphosites with B55α overexpression 293-Clone 6

Phosphosite hits inhibiting B55α in lysates (Kruse, et al 2020)

**C**

| Accession ID | Gene Name | Amino Acid Position | Inducible Flag-B55α HEK293 Clone 5 Log2 Fold Change (Dox+:Dox-) | Inducible Flag-B55α HEK293 Clone 6 Log2 Fold Change (Dox+:Dox-) | Sequence window |
|---|---|---|---|---|---|
| **Q96E09** | FAM122A | S189 | ND | -0.736343246 | PIPSPTTRFTTRRSQSPINCIRPSVLGPLKR |
| **Q96E09** | FAM122A | S76 | -1.583495331 | -1.214145049 | NSTTFPSRHGLLLPASPVRMHSSRLHQIKQE |
| **Q96E09** | FAM122A | S35 | ND | -1.394603431 | PAEGGGSGGGGGLRRSNSAPLIHGLSDTSPV |
| **Q96E09** | FAM122A | S143 | -1.51612515 | -1.505256731 | VEKSASPKRIDFIPVSPAPSPTRGIGKQCFS |
| **Q96E09** | FAM122A | S147 | -0.637639418 | -1.505256731 | ASPKRIDFIPVSPAPSPTRGIGKQCFSPSLQ |
| **Q96E09** | FAM122A | S276 | ND | -1.708797569 | VSTTTDSPVSPAQAASPFIPLDELSSK |

**D**

| | | | | |
|---|---|---|---|---|
| Q96E09 | PBIR1_HUMAN | F122A | 64 | TFPSRHGLLLPASPVRMHSSRLHQIKQEEGMDLI-NRETVHEREVQTAMQISHSWEESFS |
| Q9DB52 | PBIR1_MOUSE | F122A | 61 | TFPSRHGLLLPASPVRMHSSRLHQIKQEEGMDLI-NRETVHEREVQTAMQISHSWEESFS |
| Q6AYT4 | PBIR1_RAT | F122A | 63 | TFPSRHGLLLPASPVRMHSSRLHQIKQEEGMDLI-NRETVHEREVQTAMQISHSWEESFS |
| Q5ZLN7 | PBIR1_CHICK | F122A | 58 | TVMNRHSLFVPSSPIRIPSSRLHQIKQEEGMNLM-NRETVHEREVQAMQMSQSWEESLS |
| Q6GQ07 | Q6GQ07_XENLA | F122B | 53 | TVVNRQSLVVPSSPIRISSSRLHQIKQEEGVDLMINRETAHEREVQAMQMSQSWEESLN |
| F1QHD1 | F1QHD1_DANRE | F122B | 61 | TVVRP--NVVPSSPVRVPSTRLQRIKQEEGVDVM-NRETAHEREVQAAMQMSQSWEESLS |
| Q9VM36 | Q9VM36_DROME | F122B | 132 | SYSPLTT--AANGATLCLTPRVSQLKQEECADLN-SREVNHEREVHREIQISQSWEDLTL |
| Q20490 | Q20490_CAEEL | F122B | 162 | K-S-AEDK-FVGRGIDAPRGRIANIRRESSCS--VDSEAAHERLTKASQQVSTGFDDIAL |

**Fig. 1 | Strategy for identification of FAM122A as a SLiM containing protein and its conservation in bilateral animals. A** Steps in the pipeline towards the identification of PP2A/B55α substrates and other binders that use the SLiM. A generalized groove SLiM was derived from the similarity in p107, p130, TAU, and MAP2 sequences. This motif was permitted to have some degeneracy and variability in position from the amino-terminal phosphosite and searched within the human proteome using ScanProsite[24,25]. This list was cross-referenced to B55-related datasets and the proteins assessed for conservation (PANTHER[52,53]), gene ontology (using GENEONOTOLOGY[54,55]), and presence within IDRs (using IUPRed2A[56]). **B** Venn diagram showing common hits among the ScanProsite search and proteins in the indicated datasets (Supplementary Data 1 and Data 2). **C** Phosphosites downregulated upon doxycycline induction of B55α expression in HEK293-iB55α clones 5 and 6 in FAM122A, one of the common hits among the datasets. **D** The SLiM is conserved in bilateral animals: worms, flies, amphibians, birds, and mammals. The SLiM is boxed. "*" (asterisk) indicates positions that have a single, fully conserved residue, ":" (colon) indicates conservation between groups of strongly similar properties; "." (period) indicates conservation between groups of weakly similar properties.

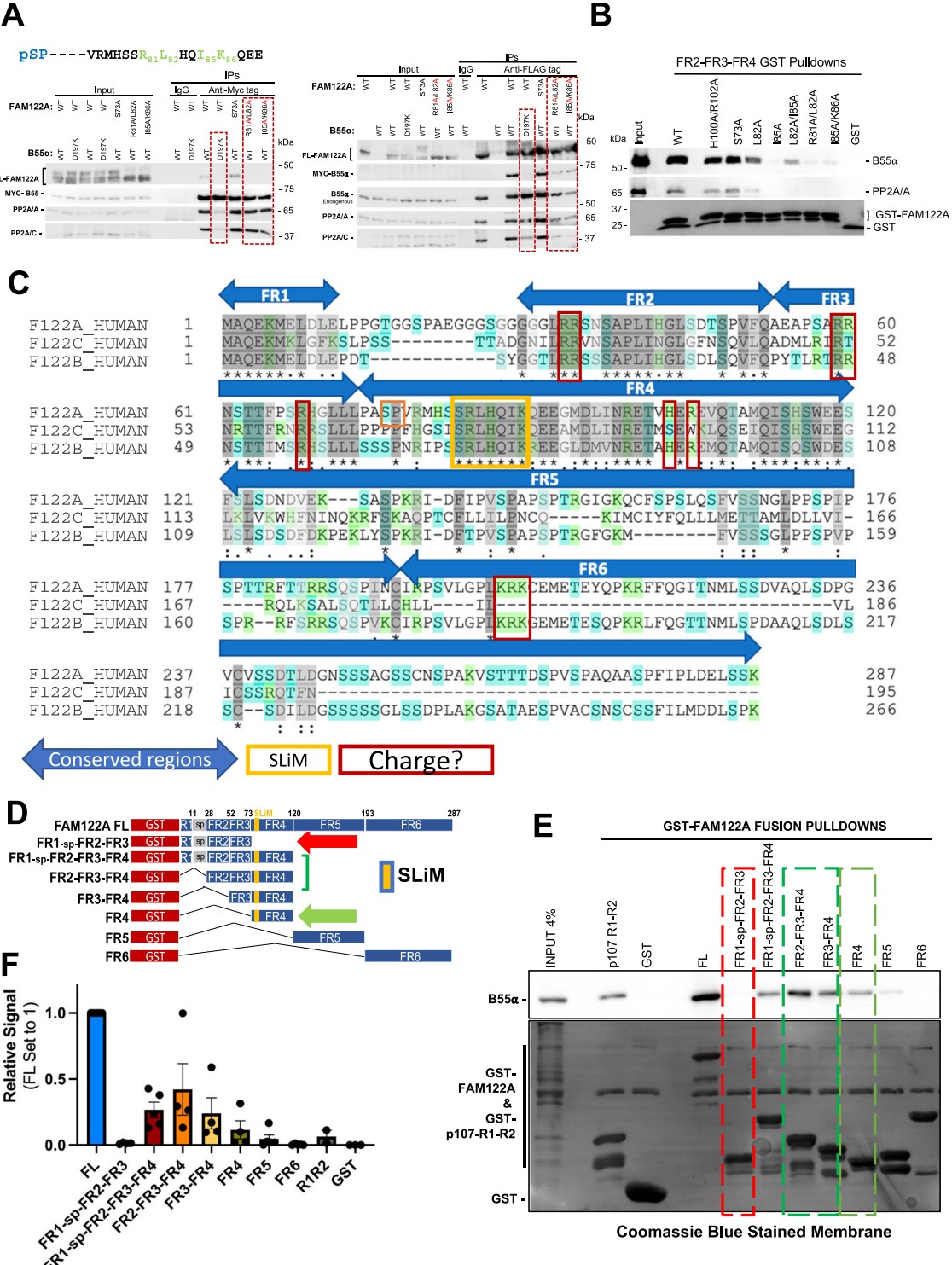

**Fig. 2 | SLiM residues are required for FAM122A binding to B55α.**
**A** HEK293T cells were cotransfected with FLAG-FAM122A and Myc-B55α WT and MTs. Anti-Myc and -FLAG IPs were analyzed by western blot (*n* = 3). **B** Glutathione beads loaded with WT and MT GST-FAM122A deletion constructs were incubated with purified PP2A/B55α and the pulldowns were analyzed by western blot, demonstrating dependence on the FAM122A SLiM residues for B55α binding. **C** Human FAM122 family amino acid sequence alignment (FAM122A: Q96E09, FAM122C: Q6P4D5, and FAM122B: Q7Z309-3) using the UniProt sequence alignment tool. Regions were selected based on amino acid conservation; the SLiM and residues that could mediate dynamic charge-charge interactions are marked. **D**–**F** GST-FAM122A pulldown assays with the indicated constructs were done as in **B** and quantitated (**F**). The following are presented as biological replicates: FL, FR1-sp-FR2-FR3-FR4, FR5, FR6 *n* = 5. FR1-sp-FR2-FR3, FR2-FR3-FR4, FR3-FR4, FR4 *n* = 4. GST *n* = 3. R1R2 *n* = 2. Data are presented as mean values +/− SEM. Source data are provided in the Source Data file.

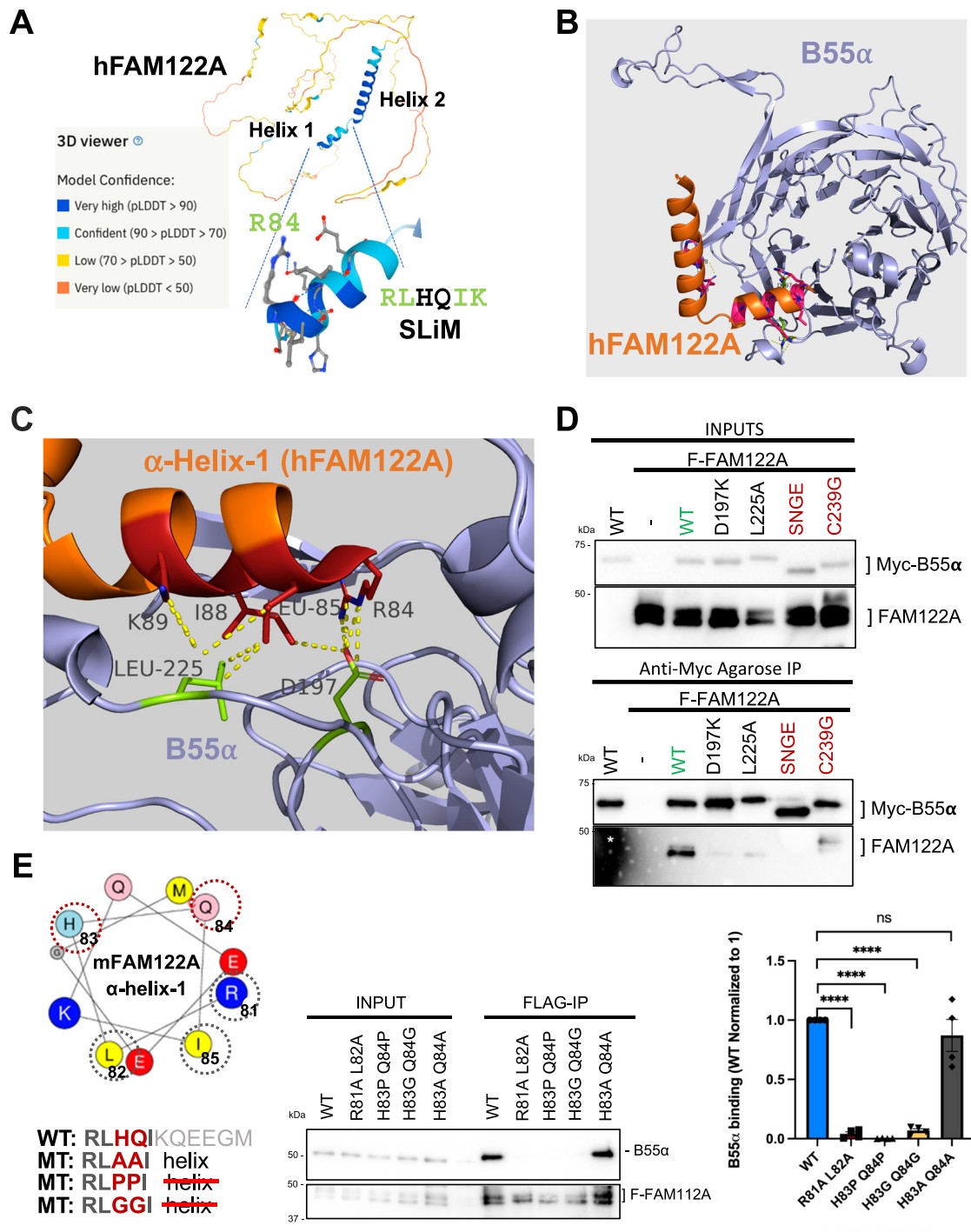

**Fig. 3 | AlphaFold2 predicts the folding of a short helix in hFAM122A containing the SLiM (SHeM), including R84 that makes contacts with D197 in B55α.**
**A** AlphaFold2 predicts the folding of two adjacent hFAM122A helices, with helix 1 containing the SLiM. **B** AlphaFold2_advanced predicts that helix one binds at the mouth of a deep groove on the top of B55α that we have previously shown interacts with B55α substrates. **C** A closeup of the model aligned to the holoenzyme shows hFAM122A R84 making contacts with D197 in B55α and L85-I88 with potential contacts to B55α L225. The position of helix-2 suggests a mechanism for blocking access to the active site and/or potential stabilization of the enzyme/inhibitor complex. Residues are shown as numbered sticks: red for hFAM122A, and lime for B55α. **D** anti-Myc IPs of HEK293T co-transfections of WT FLAG-FAM122A with Myc-

B55α and indicated mutants ($n = 3$). **E** mFAM122A α-helix-1 representation using the HeliQuest tool. The amino acids that contact B55α are marked by a grey dashed circle. The variable residues mutated to Pro or Gly to break the helix are marked by a red-dashed circle. HEK293T cells were transfected with FLAG-mFAM122A WT and helix tolerant and breaker MTs. Anti-FLAG IPs were analyzed for binding to endogenous B55α by western blot and relative binding was quantitated ($n = 4$ biological replicates). Relevant proteins are indicated. Data was analyzed for statistical significance using a One-way ANOVA (two-sided) with Dunnett correction for multiple comparisons. WT: R81A L82A $p$-value < 0.0001. WT: H83P Q84P $p$-value < 0.0001. WT:H83G Q84G $p$-value < 0.0001. WT:H83A Q84A $p$-value = 0.4076 (ns). Data are presented as mean values +/− SEM. Source data are provided in the Source Data file.

to be predominantly disordered. We then used the AF2 implementation in ColabFold[28] and AlphaFold-Multimer (AFM) to predict structures of the complexes of B55α and FAM122A and the entire complex of B55α, PP2A, the scaffold subunit, and FAM122A. Without the use of templates, AF2 in ColabFold predicted the beta-propeller folding of B55α with high accuracy compared to the experimentally determined structure (1.8 Å RMSD to PDB:3dw8, chain B). Four of five models placed the predicted short helix of FAM122A at the mouth of the B55α top groove, with the side chain of R84 forming a salt bridge (at 2.9 Å) with the side chain of B55α D197 (Fig. 3B, C, Supplementary Fig. 3A). In the model, L85 and L88 of FAM122A make hydrophobic contacts with B55α L225 (Fig. 3B, C). Consistently B55α$_{D197K}$ and B55α$_{L225A}$ fail to bind FAM122A (Fig. 3D). To demonstrate that the helical structure containing the SLiM residues is critical for binding to B55α, we determined the effect of mutations likely to disrupt/alter α-helices (Pro and Gly) vs. mutations that are typically tolerated (Ala). The mutations were introduced in positions 3 and 4 of the FAM122A SLiM (**RL**HQ**IK**), as substitution to Ala in the p107 SLiM at these positions does not affect binding. AF2 predictions were consistent with this idea, as the **RL**AA**IK** FAM122A variant formed a helix with "pLDDT" accuracy values comparable to WT FAM122A (Supplementary Fig. 3B). As predicted, mutation to Ala (**RL**AA**IK**) had no effect on binding (Fig. 3E). In contrast, FLAG-FAM122A variants with α-helix breaker residues (RLPPIK) or (RLGGIK) failed to bind Myc-B55α in anti-FLAG immunoprecipitations of lysates of HEK293T cells (Fig. 3E). Consistently, AF2 either does not bind these peptides to B55α or scores them poorly (Supplementary Fig. 3B).

## FAM122A helix-2 inhibits substrate dephosphorylation through predicted interactions of hFAM12A E100 and E104 with the active site of PP2A/C

Of note, the C-terminal long helix is predicted to be placed at a ~90° angle (Fig. 3B). Superposition of the heterodimer of B55α and FAM122A with the experimental structure of the PP2A-B55α-Scaffold complex (PDB 3dw8[13]) suggested that it could fill the space between B55α and the active site of PP2A/C (Fig. 4A, only model 1 is shown). This strongly suggests that FAM122A would block substrate access to the PP2A/C active site. This was confirmed when a retrained AFM v2.3 was made available in December 2022. AFM v2.3 produced a model of the full tetrameric complex containing FAM122A (Fig. 4B, Supplementary Fig. 4A). A complete surface model of the PP2A/B55α holoenzyme bound to FAM122A predicts that FAM122A fills the space between the active site of PP2AC and B55α with contacts to both proteins (Fig. 4B). Moreover, residues within the long helix are predicted to be in close proximity with PP2A/C residues, suggesting potential additional contacts that could stabilize holoenzyme/inhibitor binding. To gain insight into this possibility, we compared FLAG-FAM122A binding to wildtype (WT) Myc-B55α or a point mutant (C239G) that dramatically reduces B55α binding to the PP2A/A scaffold, rendering B55α monomeric in cells, yet capable of binding p107 and potential B55α substrates (Supplementary Fig. 4B). This mutation reduced binding to FAM122A and resulted in a shift in mobility that could indicate changes in phosphorylation (Fig. 3D). In addition, immunoprecipitation from lysates of cells transfected with this B55α mutant vs WT followed by mass spec shows FAM122A as the most enriched holoenzyme binder (Fig. 4C, Supplementary Data 3). This indicates that FAM122A makes contacts with other subunits in the holoenzyme. Figure 4D shows that hFAM122A E100 and E104 are predicted to make contacts with the manganese ions in the active site and the completely conserved residues in PP2A/C that coordinate substrate phosphate (R89, H118, and R214)[9]. Consistently, a mFAM122A variant with the residues corresponding to hFAM122A E100 and E104 mutated to Ala or Pro, lack inhibitory activity towards the pDiFMUP substrate (Fig. 4E), but retain binding to B55α through helix1, even when the corresponding E residues in mFAM122A are changed to a positive charge (E to K reversals,

Supplementary Fig. 4C). Altogether these data show that helix-2 acts as the inhibitory helix, whereas helix-1 mediates FAM122A binding to the holoenzyme through contacts in the B55α top groove.

## FAM122A abrogates p107 binding to B55α in vitro and in cells

A previous report suggested that FAM122A inhibits PP2A/B55α by a mechanism involving degradation of the PP2A/C catalytic subunit[22]. However, we have not seen any effect in the catalytic subunit of the B55α holoenzyme when we co-express FAM122A in cells (Fig. 2A). In contrast, we have observed that under comparable concentrations of FAM122A and p107, FAM122A pulls down more B55α in vitro (Fig. 2E, Supplementary Fig. 2B), suggesting comparatively higher affinity. In agreement with this, purified HA-FAM122A strongly inhibits B55α:p107 binding in vitro at FAM122A concentrations comparable to the PP2A/B55α complex (nM range) and with vast excess of p107$_{585-691}$ (μM range) (Fig. 5A, B) in in vitro competition assays. Additionally, a dose-dependent increase in transfected FLAG-FAM122A in HEK293T cells displaces FLAG-p107 from Myc-B55α (Fig. 5C). At comparable FLAG antibody signals, FAM122A abolishes p107 binding, which is consistent with FAM122A using its SLiM to block substrate binding and other conserved residues to bind with higher affinity than B55α substrates (among all PP2A/B55α substrates tested by us, p107 binds the tightest to B55α). Further supporting a SLiM competition model, AF2 also predicts a helix spanning the p107 **R**$_{621}$**VKEVR** SLiM sequence, which prompted our use AFM[29] to predict the interaction of p107 residues 612-648 with B55α. The original AlphaFold-Multimer was trained on multi-protein complexes in the Protein Data Bank deposited before April 2018. The model shows a helix formed from residues 620-633 which exactly superimposes the SLiM of p107 with that of FAM122A, with interaction of R621 with D197 and V625 with L225 (Fig. 5D). This binding model is compatible with the placement of p107 P-S640, located C-terminally with respect to the SLiM, at the PP2A/C active site. We therefore sought to determine if dephosphorylation of S640 was, like S615, dependent on the SLiM using p107-R1-R2 mutant variants that only contain a single SP-phosphosite. To make the phospho-S640 recognizable by an anti-p-SP antibody, we mutated the residue preceding S640 to Met. Figure 5E shows that S640 is dephosphorylated with kinetics comparable to S615, and that dephosphorylation of both phosphosites is dependent on the SLiM. Figure 5F schematically summarizes the FAM122A competitive mechanism of inhibition. Finally, we determined the approximate affinity of FAM122A and p107-R1-R2 for B55α at the interaction equilibrium using a previously described depletion assay[30]. The K$_D$ for GST-FAM122A/B55α was ~2.44 fold lower than that of GST-p107-R1-R2:B55α (Supplementary Fig. 5A, B), which is consistent with FAM122A effectively competing with p107 for binding to B55α in vitro and in cells.

## FAM122A promotes proliferation

We next investigated the functional role of FAM122A in the cell cycle using CRISPR knockouts. FAM122A knockout in HEK293 cells dramatically reduces colony formation and proliferation (Fig. 6A–C). Inhibition of proliferation was also observed in T98G and U-2 OS cells using 2 different sgRNAs (Fig. 6C, Supplementary Fig. 6A). Reconstitution of GFP-FAM122A in HEK293-FAM122A-KO cells partially rescued the proliferation defect in a SLiM-dependent manner (Fig. 6D, Supplementary Fig. 6B), indicating that the defects on FAM122A are at least partially dependent on PP2A/B55α inactivation. This is further supported as FAM122A knockout in HEK293 cells results in increased phosphatase activity relative to wildtype control in dephosphorylation of substrates phosphorylated on S/T-P sites (Supplementary Fig. 6C).

## FAM122A is required for timely cell cycle entry and progression through the G1 phase of the cell cycle following mitogen stimulation

Because PP2A/B55α opposes CDK inactivation of the pocket proteins, pRB, p107, and p130[17–19,31], and we have shown that PP2A/B55α directly

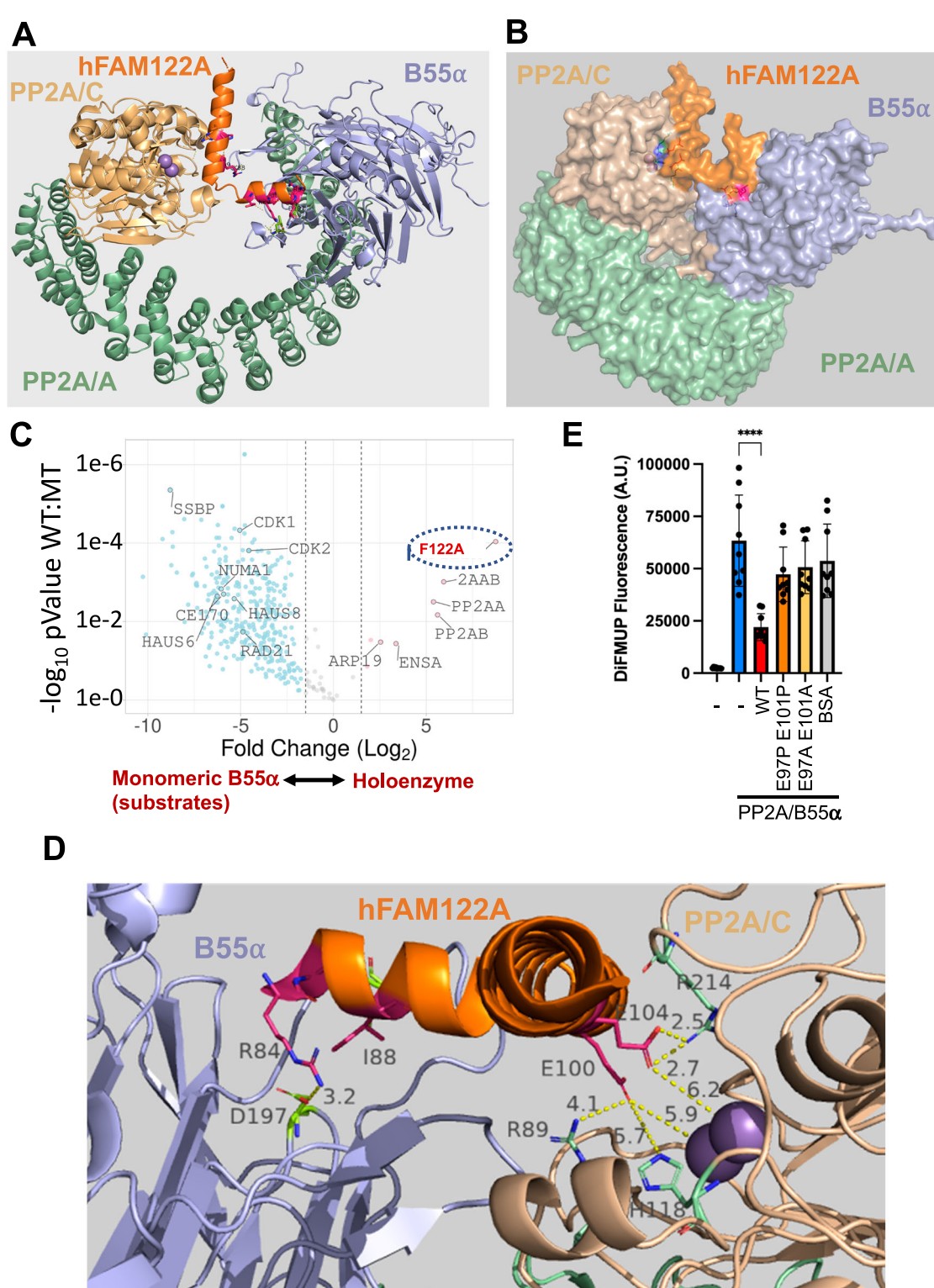

dephosphorylates p107 in vitro and in cells[14], we determined if knockout of FAM122A had any effects in cell cycle re-entry and progression through the cell cycle of T98G cells in response to serum stimulation. Parental and FAM122A KO T98G cells were serum starved for 3 days and then stimulated with serum. Cells were collected at the indicated points for DNA content FACS cell cycle and western blot analysis. FACS analysis shows a >4 h delay in the cells progressing from G0 to mitosis (Fig. 7A, nocodazole was added 16 h post-release to prevent cells from entering the next cycle). To observe the delay more

clearly, we performed a serum starvation and re-stimulation experiment where cells were collected hourly from 16 to 20 h. By 20 h the KO cells had not reached the number of cells in S phase already seen in the parental cells by 16 h (Fig. 7B and Supplementary Fig. 7B). Moreover, double staining with EdU and PI, showed an increase in the number of cells remaining in G1 at 12, 15 and 18 h. post serum stimulation in WT (65.3%, 38.6%, 26%) vs. KO cells (69.7%, 51.9%, 39%) (Fig. 7C, quantitated in Supplementary Fig. 7A). Western blot analysis showed a delay in the upregulation and reduced peak levels of Cyclin D1 that are

**Fig. 4 | Helix-2 of FAM122A interacts with the active site of PP2A/C and inhibits the enzyme. A** Superimposition of the AF2 B55α:hFAM122A model aligned to the structure of the holoenzyme (3DW8) using the B55α subunits in each structure positioning helix-2 of hFAM122A near the active site of PP2A/C. **B** AlphaFold-Multimer v2.3 Surface model of the PP2A/B55α holoenzyme bound to hFAM122A$_{76-122}$. **C** Pulldown IP mass spec analysis of Myc-B55α-C239G (renders B55α monomeric) vs Myc-B55α-wildtype (holoenzyme) demonstrating FAM122A preferentially binds the holoenzyme. Statistical significance was determined using a two-sided student's *t*-test (no multiple comparisons). Other relevant proteins are indicated (Supplementary Data 3). **D** AlphaFold-Multimer v2.3 model closeup of hFAM122A and the three subunits of the PP2A/B55α holoenzyme. Distances for the predicted

contacts between residues in hFAM122A with B55α and PP2A/C are indicated. The distance of E100 and E104 to the Mn$^{++}$ metals is also indicated. **E** DiFMUP phosphatase assay (15 min duration) of purified PP2A/B55α holoenzyme preincubated with purified full-length HA-tagged wild-type or helix-2 FAM122A mutants of residues predicted to promote holoenzyme inhibition. Each reaction was performed in triplicate with a $n = 3$ and statistical significance analyzed using a One-way ANOVA (two-sided) with Dunnett correction for multiple comparisons. PP2A/B55: + FAM122A WT *p*-value < 0.0001. PP2A/B55: + FAM122A E97P E101P *p*-value = 0.0706 (ns). PP2A/B55: + FAM122A E97A E101A *p*-value = 0.2067 (ns) PP2A/B55 + BSA *p*-value = 0.4483 (ns). Data are presented as mean values +/− SD. Source data are provided in the Source Data file.

consistent with the delays in pRB phosphorylation (a PP2/B55α substrate[7,14,32]), the expression of E2F-dependent gene products (p107, Cyclin A) and the degradation of p130, which is dependent on CDK2 activity (Fig. 7D). Similar delays in cell cycle progression and the expression of these markers were observed in cells where FAM122A was knockdown by siRNA (Supplementary Fig. 7C, D).

Because the decreased expression of cyclin D1 could explain a delay on pRB inactivation and passage to the restriction point and cyclin D1 expression is controlled by early mitogenic signaling[33], we performed a serum starvation and re-stimulation experiment collecting cells at short time points preceding the expression of cyclin D1. Lysates of these cells revealed delays in the activation of ERKs and AKT, both upstream regulators of cyclin D1 (Fig. 7E). The magnitude of peak signaling was also clearly lower (Fig. 7E). ERKs control cyclin D1 transcription, while AKT controls the stability of Cyclin D1 by inactivating GSK3β, which is known to promote Cyclin D1 degradation[33]. Consistently, inactivation of GSK3β is delayed in FAM122A-KO cells (Fig. 7E). These results, together with the FAM122A SLiM-dependent defects in proliferation (Fig. 6), strongly indicate that FAM122A controls PP2A/B55α activities that attenuate early mitogenic signaling including AKT T308[34], a site known to be directly regulated by PP2A/B55α, as well as other direct/indirect effects that oppose ERK activation (Supplementary Fig. 7E).

## FAM122A is required for checkpoint activation in response to replication stress (RS)

Our data together with that published by others supports that FAM122A is an abundant regulated inhibitor of PP2A/B55α[3]. FAM122A has been recently shown to be phosphorylated and inhibited by CHK1 in NSCLC cell lines promoting activation of PP2A/B55α, which stabilizes WEE1 leading to inhibition of CDK1[23]. Interestingly, ablation of FAM122A restores WEE1 expression and the basal levels of inactive CDK1, restores fork replication speed, and dramatically reduces RS and DNA damage, suggesting that maintaining downstream control of CDK1 through FAM122A is the main mechanism by which CHK1 ensures genomic stability in these cells[23]. However, our data are only partially consistent with this model. Analysis of cell cycle effects of ablation of FAM122A in HEK293 cells using BrdU incorporation and propidium staining assays shows a clear reduction in the incorporation of BrdU to DNA during S phase, without a decrease in the number of cells with PI staining corresponding to S phase cells (Fig. 8A, the mean fluorescence intensity is 4181 for the CRISPR control and 1662, for the FAM122A KO, $n = 3$). This suggests that cells are synthesizing DNA more slowly and that the decreased proliferation in these cells, which have a defective pRB pathway, could be the result of extension of S-phase length, rather than accumulation of cells in G1. Thus, we determined the effect of FAM122A KO in DNA-fiber length in exponentially growing HEK293 cells. DNA fiber length synthesis was decreased by -18.8%, as determined by IdU and CIdU incorporation assays (Fig. 8B). These results show that elimination of FAM122A induces replication stress, which was not reported in NSCLC cell lines[23]. Thus, these cells are synthesizing DNA more slowly and their decreased proliferation likely results from an extension of S-phase length.

HEK293 are synchronized by either single or double thymidine blocks (Supplementary Fig. 8A), which deplete endogenous nucleotides. In contrast, FAM122A KO HEK293 cells are refractory to synchronization at the G1/S transition, and thymidine washout resulted in massive cell death. Figure 8C shows a thymidine challenge that resulted in time-dependent accumulation of HEK293 cells at the G1/S transition and in S phase that correlated with accumulation of Cyclin E peaking at 18 h. By contrast, FAM122A-KO cells exhibited diminished sensitivity to nucleotide depletion and although they slowed down through S phase, they continued to progress through G2 and completed mitosis. Completion of mitosis in FAM122A KO cells was demonstrated by adding nocodazole to replicate samples 12 hours post washout, which resulted in massive accumulation of cells in G2/M with very few cells remaining in S phase by 24 h (Fig. 8C, note gray overlay over the untreated KO cells). This was followed by massive cell death (Sub-G1 DNA-accumulation). Progression of KO cells though S/G2/M was also clear from the accumulation of cyclins A and B that peaked at 18 and 24 h respectively. We also noted time dependent accumulation of p53, which started as cells were moving to G2/M and was followed by accumulation of p21 in G1 (p21 levels increase at 24 and 32 h, but this increase is blocked by nocodazole, indicating that p21 expression is upregulated in late mitosis or G1 (Fig. 8C). To confirm the diminished sensitivity to nucleotide depletion we used hydroxyurea (HU), which results in rapid and complete inhibition of DNA synthesis at a 2 mM concentration. Both the parental and FAM122A KO cells arrest in S-phase due to rapid nucleotide depletion, as the G1 centered peak moves to the right of the G1/S border delimited by a dashed red line. However, while WT cells arrest in S phase and are viable, the FAM122A KO cells die. Since the cells in the presence of HU are either at the G1/S border unable to initiate DNA synthesis or arrested in S phase, the FAM122A KO PI/FACS data shows that these S phase arrested cells are unable to survive. Consistent with these findings, the G1/S and intra-S phase arrest in parental HEK293 cells resulted in rapid check point activation as demonstrated by activation of CHK2 (P-T68) and CHK1 (P-S345) and time-dependent accumulation of cyclin E in parental cells. Checkpoint activation was largely attenuated in FAM122A-KO cells, which exhibited rapid and time-dependent accumulation of γH2AX, sub-G1 DNA, and potent attenuation of both CHK2 and CHK1 activation (Fig. 8D). Of note, we did not observe changes in WEE1 expression, indicating that the intra-S phase arrest mediated by FAM122A upon nucleotide deprivation is likely independent of WEE1. Altogether, these data strongly suggest that FAM122A controls PP2A/B55α-dependent steps upstream of these kinases. This is consistent with the ability of PP2A/B55α to negatively regulate ATM[21], but likely involves other factors/substrates in this pathway (Supplementary Fig. 8G).

We next determined if the abrogation of the thymidine arrest checkpoint in HEK293 FAM122A KO cells could be rescued by reconstitution of FAM122A. Figure 8E shows that HEK293 FAM122A KO cells fail to arrest in S phase following 24 h incubation with 2 mM thymidine. Reconstitution of WT FAM122A, but not FAM122A SLiM-mutant, restored the arrest in S phase (Fig. 8E and Supplementary Fig. 8I).

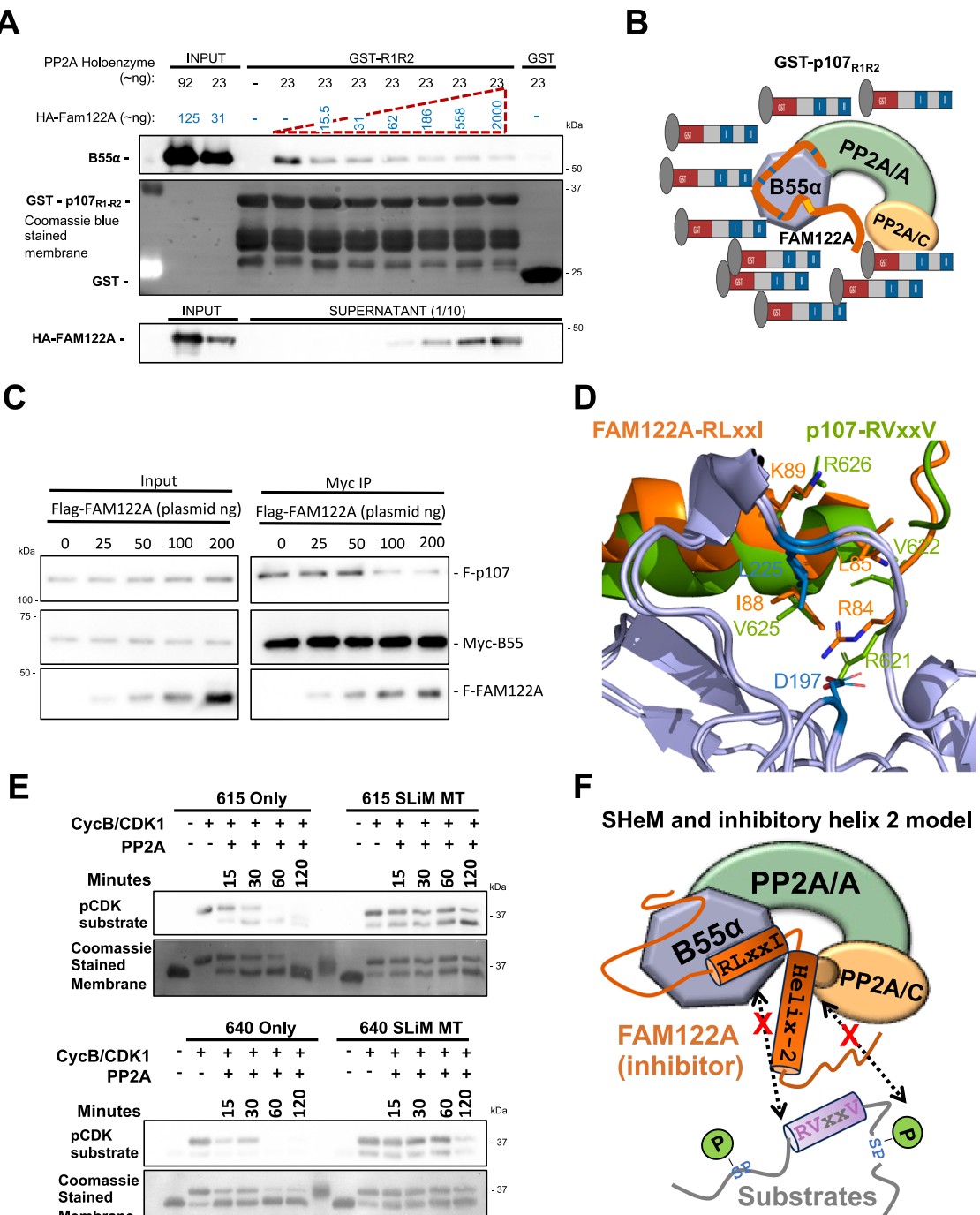

**Fig. 5 | FAM122A effectively competes binding to the holoenzyme, which is consistent with the AFM model predicting superimposition of the FAM122A and p107 SHeMs. A** Purified PP2A/B55α holoenzymes were incubated with purified HA-FAM122A (expressed in *E coli*) starting with the same concentration as B55α in each sample up to a ~100X. B55α: R1R2 binding in GST-p107-R1R2 pull-downs was inhibited at the lowest FAM122A concentration. (*n* = 3). Source data are provided in the Source Data file. **B** Schematic interpretation of **A. C** Co-transfection of invariable amounts of Myc-B55α and FLAG-p107 plasmids with increased amounts of FAM122A, as indicated. (*n* = 3). Source data are provided in the Source Data file. Anti-Myc immunoprecipitates were analyzed by western blot and relevant proteins are indicated. **D** AlphaFold-Multimer model of superimposing a predicted helix containing the p107 with the predicted helix-1 of hFAM122A. Conserved SLiM residues in both proteins superimpose and make contacts with D197 and L225. Residue numbering corresponds to the human proteins. **E** GST-R1R2 constructs created with a single phosphosite (S615 or S640). Arginine to Alanine mutations disrupted the SLiM motif in the indicated constructs. Western blot visualization of time-dependent kinase-dephosphorylation assays of the R1R2 constructs. GST-R1R2 variant beads were phosphorylated with CDK1/cyclin B kinase. Washed phosphorylated beads were incubated with PP2A/B55α for the indicated times (*n* = 3). **F** Schematic representation of the substrate-competitive SHeM mechanism of inhibition, which involves the two helices. Helix-1 (SHeM) mediates binding to B55α and helix-2 acts as inhibitory flapper by directly contacting the active site.

It has been recently reported that inhibition of CHK1 reduces phosphorylation of FAM122A presumably on S37, increases binding to B55α and reduces binding of 14-3-3 proteins, which results in FAM122A upregulation in the nuclear fraction of A549 lung cancer cells[23].

FAM122A is also phosphorylated in cells at multiple sites that are consensus for the CDKs and ERK[3], indicating a potential regulation during the cell cycle. Therefore, we determined if FAM122A interaction with B55α is regulated during the cell cycle or in response to

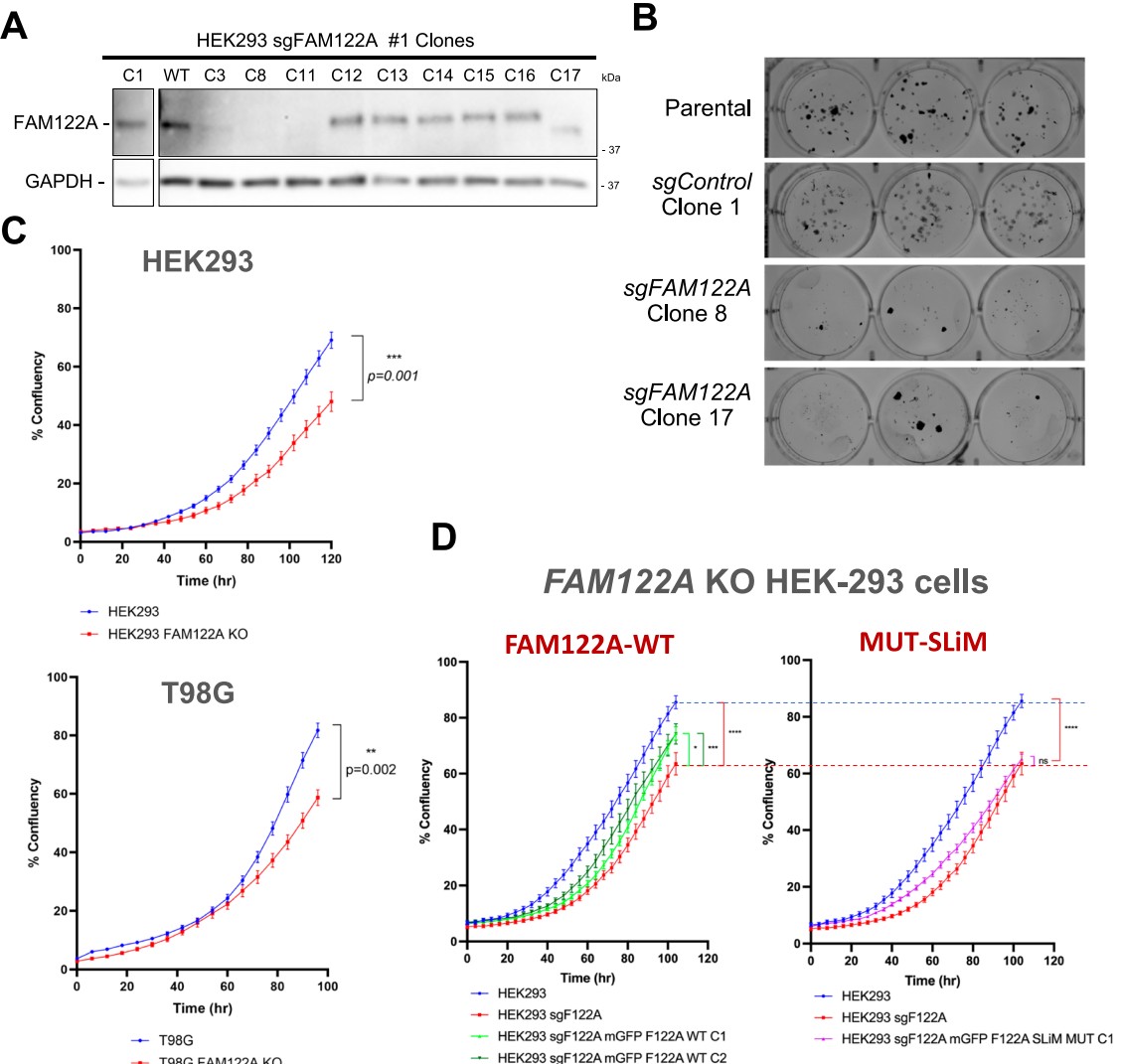

**Fig. 6 | Ablation of FAM122A results in reduced proliferation that is rescued by re-expression of FAM122A in a SLiM-dependent manner. A** Western blot analysis of FAM122A CRISPR HEK293 clones (the C8 KO and C17 truncation were selected for further analysis) (*n* = 3) **B** Colony formation assays of control (parental and C1) and KOs (C8 and C17). **C** Ablation of FAM122A in HEK293, T98G cells inhibits proliferation. Cells were seeded in triplicate and the percent confluence was imaged with an Incucyte SX5. Statistical analyses of proliferation curves comparing parental cells to FAM122A knockout cell lines was conducted using a Wilcoxon matched-pairs signed rank test (two-sided). Data are presented as mean values +/− SEM. **D** A cassette directing the expression of FAM122A WT and SLiM-MT was stably introduced in HEK293 FAM122A KO cells. Expression of FAM122A wildtype but not SLiM mutant rescued the proliferation defect. Statistical analyses of proliferation curves comparing rescues to FAM122A knockout cell lines was conducted using a Friedman's test (two-sided) of the last 10 measurements. HEK293 vs KO *p* value < 0.0001 (****), HEK293 WT Clone 1 *p* value = 0.0436 (*), HEK293 WT Clone 2 *p* value = 0.0002 (***), HEK293 SLiM MUT Clone 1 *p* value = 0.2640 (ns). FAM122A WT and SLiM mutant reconstitution are show in separate graphs compared to the same controls to more clearly visualize partial rescue (green curves) vs. absence of rescue with the SLiM mutant (magenta curve). Data are presented as mean values +/− SEM. Source data are provided as a Source Data file.

checkpoint activation. We did not observe changes in the levels of B55α bound to tagged FAM122A when comparing exponentially growing HEK293 and T98G cells, to cells arrested at the G1/S transition with thymidine, the G2/M transition with the CDK1 inhibitor (RO3306), and in pseudo-metaphase with nocodazole (Supplementary Fig. 8B, C). Also, we did not observe regulation of these complexes in T98G cells progressing through the cell cycle from quiescence or exiting mitosis and progressing through G1 from a nocodazole arrest (Supplementary Fig. 8D, E). These data suggest that the overall amount of FAM122A/B55α complex is constant under all these conditions and that regulation may depend on the colocalization of the complex and substrates and dynamic relative changes in FAM122A/substrate affinity for B55α.

We also determined if HU and/or inhibition of CHK1, which cause strong replication stress promote changes in the FAM122A/B55α interaction and binding to 14-3-3 proteins, which have been proposed

to sequester FAM122A in response CHK1 phosphorylation[23]. We did not detect changes in the B55α/FAM122A interaction in GFP-FAM122A transfected cells (Supplementary Fig. 8F). Surprisingly, we also did not detect formation of a FAM122A/14-3-3 protein complex upon induction of replication stress (Supplementary Fig. 8F). Therefore, FAM122A is critical to establish checkpoints in response to replication stress that modulate CHK1 and CHK2 signaling.

Given the apparent lack of regulation of the FAM122A/B55α interaction as determined via immunoprecipitation, we determined the localization of FAM122A during the cell cycle. Transiently co-transfected RFP-FAM122A with EGFP-h-H2B in HEK293T cells revealed that FAM122A is largely, if not exclusively, localized in the nucleus in interphase, and it is released in the cytoplasm in prometaphase following nuclear envelop breakdown (NEB) (Fig. 9A). Co-transfection of RFP-FAM122A, BFP-B55α and EGFP-h-H2B in 293 T cells reveals that a

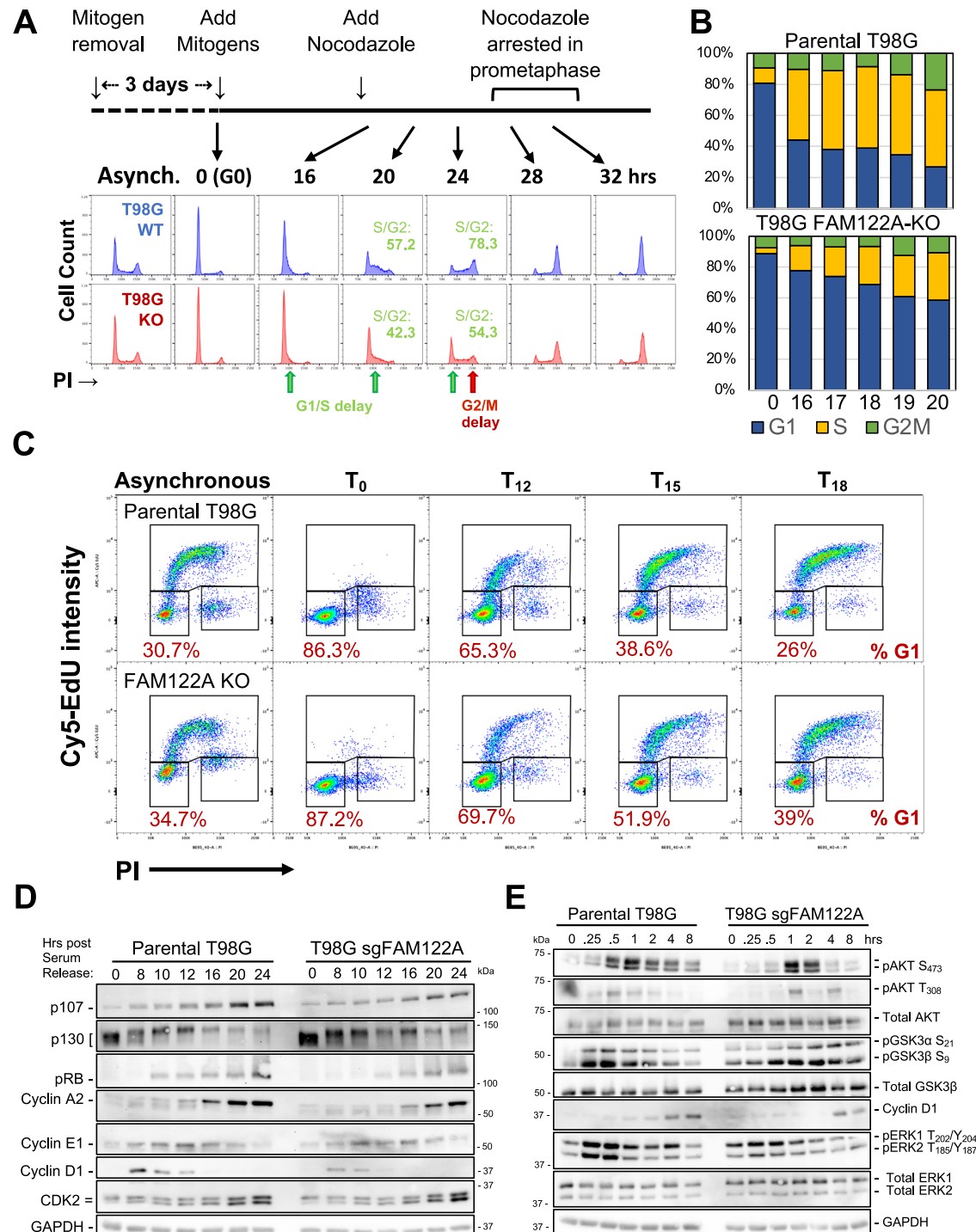

**Fig. 7 | FAM122A is required for cell cycle entry and progression through the G1 phase of the cell cycle following mitogen stimulation. A** FAM122A-WT and -KO T98G cells were serum starved for 60 h and re-stimulated with DMEM supplemented with 10% FBS and collected at the indicated time points and analyzed by PI/ FACS. Nocodazole was added to prevent cells from progressing beyond mitosis. A >4 h delay is observed by the time the cells reach mitosis, but the delay is already noticeable at the G1/S border. **B** A serum starvation-restimulation experiment focused on the G1/S transition (1 h time points) that shows a delay in cells entering in S-phase. **C** T98G Parental and KO cells were serum-starved for 72 h and subject to serum-restimulation. 10 μM EdU was added 1 h prior to collection. The cells were fixed with 4% PFA in PBS, washed, and subject to saponin-based permeabilization

followed by the Cy5-azide click-chemistry reaction and stained with PI/FACS analysis. The percentage of cells in G1 is indicated, suggesting delays prior to the onset of S-phase. **D** Delays in the expected modulation of pRB proteins and the expression of E2F-depedent gene products (Cyclins E and A and p107). The earliest defect is the limited expression of cyclin D1, whose expression is regulated by mitogens. Experiment performed in biological triplicate. **E** The effects of FAM122A KO in the activation/inactivation of key components of the major mitogenic signaling pathways regulating cyclin D1 expression in serum-starved and restimulated T98G during early signaling (AKT, ERKs, GSK3β and their relevant phosphoforms are indicated). Experiment performed in biological triplicate. Source data are provided as a Source Data file.

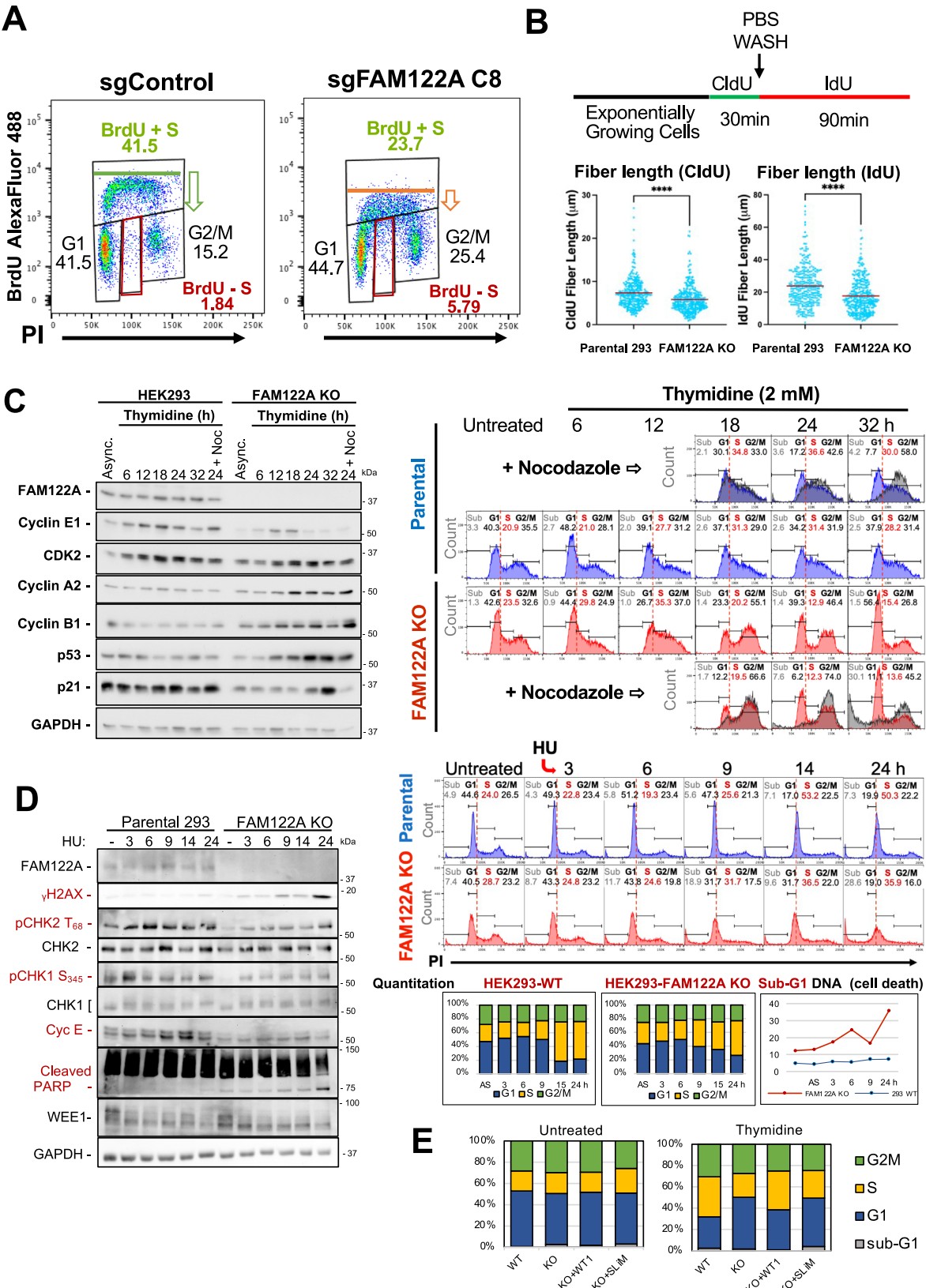

fraction of B55α localizes in the nucleus and that both FAM122A and B55α are excluded from condensed chromosomes in mitosis (Fig. 9B). To demonstrate that B55α and FAM122A form a physical complex in the nucleus we used a Split-FAST approach[35]. B55α-N-FAST and FAM122A-C-FAST10 vectors were co-transfected in U-2 OS cells and the FAM122A/B55α nuclear complex was detected upon incubation with

the HMBR fluorogen (Fig. 9C). Moreover, we used immunofluorescence to determine the localization of both FAM122A and B55α in the interphase and mitosis of U-2 OS cells. As in HEK293T cells, FAM122A is nuclear in interphase, but it is not bound to chromatin in mitosis. A large fraction of B55α is also found in the interphase nucleus, and not bound to chromatin upon NEB. Of note, in late anaphase, both

**Fig. 8 | Ablation of FAM122A abrogates G1/S checkpoint activation in response to nucleotide depletion, which causes RS. A** Ablation of FAM122A KO in HEK293 cell results in a decrease in the overall incorporation of BrdU in cells in S phase and the appearance of a fraction of cells arrested in S-phase (BrdU negative, red boxes). **B** HEK293 Parental and FAM122A KO cells were subjected to CldU incorporation, washout, and IdU incorporation to measure DNA fiber length. FAM122A KO cell fiber length was shorter, demonstrating impaired DNA synthesis. Statistical significance was determined by Mann-Whitney U Test (non-parametric two-sided). Analyzed fibers were the result of three independent replicates. CldU Parental: KO *p*-value < 0.0001. IdU Parental:KO *p*-value < 0.0001. **C** HEK293 cells were treated with 2 mM thymidine for the indicated times and analyzed by PI flow and western

blot using the indicated specific antibodies. These data demonstrate that FAM122A KO cells are refractory to arrest under thymidine nucleotide depletion at the G1/S border and in S-phase. Experiment performed in biological triplicate. **D** HEK293 cells were treated with 2 mM HU as indicated analyzed as in **B**. Attenuation of CHK2 and CHK1 phosphorylation indicates an ineffective checkpoint, which results in increased DNA damage (γH2AX accumulation) and cell death (increased PARP cleavage and increased sub-G1 DNA content). Experiment performed in biological triplicate. **E** Parental HEK293, FAM122A KO, and GFP-FAM122A WT and SLiM-MT reconstituted cells were challenged with 2 mM Thymidine for 24 h to initiate an S-phase arrest. The arrest was rescued in the WT-FAM122A expressing cells but not in SLiM-MT-FAM122A reconstituted cells.

FAM122A and B55α appear to be recruited to chromatin (Supplementary Fig. 9A). Consistent with the nuclear localization of FAM122A, we identified a conserved nuclear localization signal (NLS, $_{201}$RKK), that upon mutation ($_{201}$RKK to AAK) prevented FAM122A from reaching the nucleus (Supplementary Fig. 9B). Altogether these data show that FAM122A and B55α interact in the nucleus and that during mitosis, chromatin is devoid of FAM122A and B55α.

Given the apparent lack of global regulation of the B55α/PP2A complex in response to replication stress, we tested the hypothesis that FAM122A localization to chromatin is regulated in response to replication stress. HEK293 FAM122A-KO with and without FAM122A reconstitution were treated with HU for 24 hours in glass chambers and soluble nuclear proteins were extracted with cytoskeletal buffer[36]. DMSO-treated cells exhibit occasional γ-H2AX foci likely due to endogenous replication stress. Treatment with HU results in extensive DNA damage in the absence of FAM122A. Reconstitution of FAM122A in untreated cells results in clear localization of FAM122A at the chromatin and the larger foci colocalize with γ-H2AX. Treatment with HU resulted in strong recruitment of FAM122A to the sites of DNA damage (Fig. 9D). We next determined the effect of FAM122A reconstitution in γH2AX in HEK293 FAM122A-KO cells treated with HU for 3 h. As expected, FAM122A suppressed the γH2AX signal in a SLiM-dependent manner (Fig. 9E, Supplementary Fig. 9C). Therefore, these data shows that FAM122A subnuclear localization is regulated and that the relatively small fraction of FAM122A that is recruited to DNA-damage foci might be critical for the establishment of the checkpoint and suppression of DNA damage accumulation.

## Discussion

Using a degenerate version of a consensus SLiM found in PP2A/B55α substrates p107 and TAU[14], we have found that FAM122A, a previously identified inhibitor of PP2A/B55α, contains a functionally conserved SLiM sequence that is required for binding to B55α. FAM122A effectively competes with the model substrate p107 for binding to B55α in vitro and in cells. A previous report suggested that FAM122A binding to the PP2A/B55α holoenzyme results in downregulation of the catalytic subunit PP2A/C[22], but we have not observed changes in PP2A/C binding to the holoenzyme upon FAM122A upregulation. The discovery and validation of the requirement of the SLiM residues in FAM122A allows a revision of our proposed SLiM to p[ST]-P-x(4,10)-[RK]$_1$-[VL]$_2$-x$_3$-x$_4$-[VI]$_5$, but increased degeneracy at positions +2 and +5 is likely possible. As for the p107 SLiM, we find that a positively charged residue is not required at the final position for FAM122A binding to B55α. We have also shown here that the SLiM works to mediate dephosphorylation of sites both C-terminal and N-terminal from the SLiM, as both p107 S615 and S640, which flank the SLiM are dephosphorylated in a SLiM dependent manner.

We also show that the conserved SLiM is located in a region of FAM122A that is predicted by AlphaFold2 to form an α-helix, where the **RL**xx**I**-SLiM residues R$_{84}$ and L$_{85}$-I$_{88}$ mediate a salt bridge with B55α-D197 and hydrophobic contacts with B55α-L225, respectively. AlphaFold-Multimer also predicted an α-helix containing the p107

SLiM (**RV**xx**V**) interacting with the same residues on B55α, such that both the p107 and FAM122A α-helices superimpose. As SLiMs are often viewed as extended structures, it appears appropriate to refer to this motif as Short Helical Motif (SHeM). A comparable docking interaction was identified in the C-terminus of the retinoblastoma protein pRB, which forms a 21 amino acid α-helix that has core residues that form a hydrophobic face[37]. If these residues are mutated to Ala, Cyclin D/CDK4 cannot phosphorylate pRB; however, substitution of other adjacent residues to Ala, which has high helical propensity[38], had no effect. In contrast, disruption of α-helix by proline substitution, which prevents helix formation, impaired pRB dephosphorylation[37].

Our data-guided models predict that FAM122A simultaneously contacts both B55α and PP2A/C with dedicated α-helices 1 and 2. The SHeM residues in helix-1 are essential for binding. However, FAM122A E101 and E104, which are predicted to contact the manganese metal ions and the three invariable residues in all PPPs that coordinate substrate phosphate, are not essential for binding, but do inhibit dephosphorylation of the pDiFMUP substrate. This indicates that FAM122A E101 and E104 provide weak electrostatic interactions that allow helix-2 to occlude the active site preventing substrate access, but the main determinant for substrate recognition in the SHeM. Additional contacts in the long IDRs upstream and downstream of the α-helices in FAM122A are likely to contribute to binding strength and may potentially be regulated by multiple phosphorylation sites that are known to be phosphorylated in cells in a cell-cycle-dependent manner[3].

We have also noted that in the FAM122A-PP2A-holoenzyme complex, the curvature of the PP2A/A scaffold increases dramatically as compared to the PDB:3dw8 structure (Supplementary Fig. 4D). We propose three potential mechanisms that are not mutually exclusive. First, the C-terminal residues in PP2A/C are present in the construct of the catalytic domain (sequence RRGEPHVTRRTPDYFL) in PDB:3dw8 but they are not ordered. Alphafold2 is confident about the positions of those residues, which contact B55α (contacting residues 199–216 in B55α; average pLDDT score = 71.3). The C-terminal domain of PP2A/C is methylated and phosphorylation and methylation play a role in PP2A/B55α holoenzyme formation[6], so it is tempting to speculate that methylation of L$_{309}$ and/or phosphorylation of T$_{304}$, Y$_{307}$, may affect the interaction of B55α and the curvature of PP2A/A. Second, the presence of FAM122A bound to the holoenzyme may also directly affect PP2A/A curvature, bringing B55α closer to the active site. Third, the 3dw8 structure contains the cyanobacterial toxin microcystin in the active site of PP2A/C, which may increase the distance between PP2A/C and B55α, decreasing the curvature of the PP2A/A scaffold. Finally, we note that a cryo-EM structure of the tetrameric complex (PDB: 8SO0), released while this paper was under consideration, has an RMSD to our AlphaFold-Multimer model (released March 2023) of 1.01 Å over the backbone atoms of 1231 residues[39].

Ablation of FAM122A via CRISPR in multiple cell lines resulted in defects in proliferation and attenuation of the G1/S and intra-S checkpoint induced by replication stress. These defects were rescued by reconstitution of FAM122A in a SLiM dependent manner,

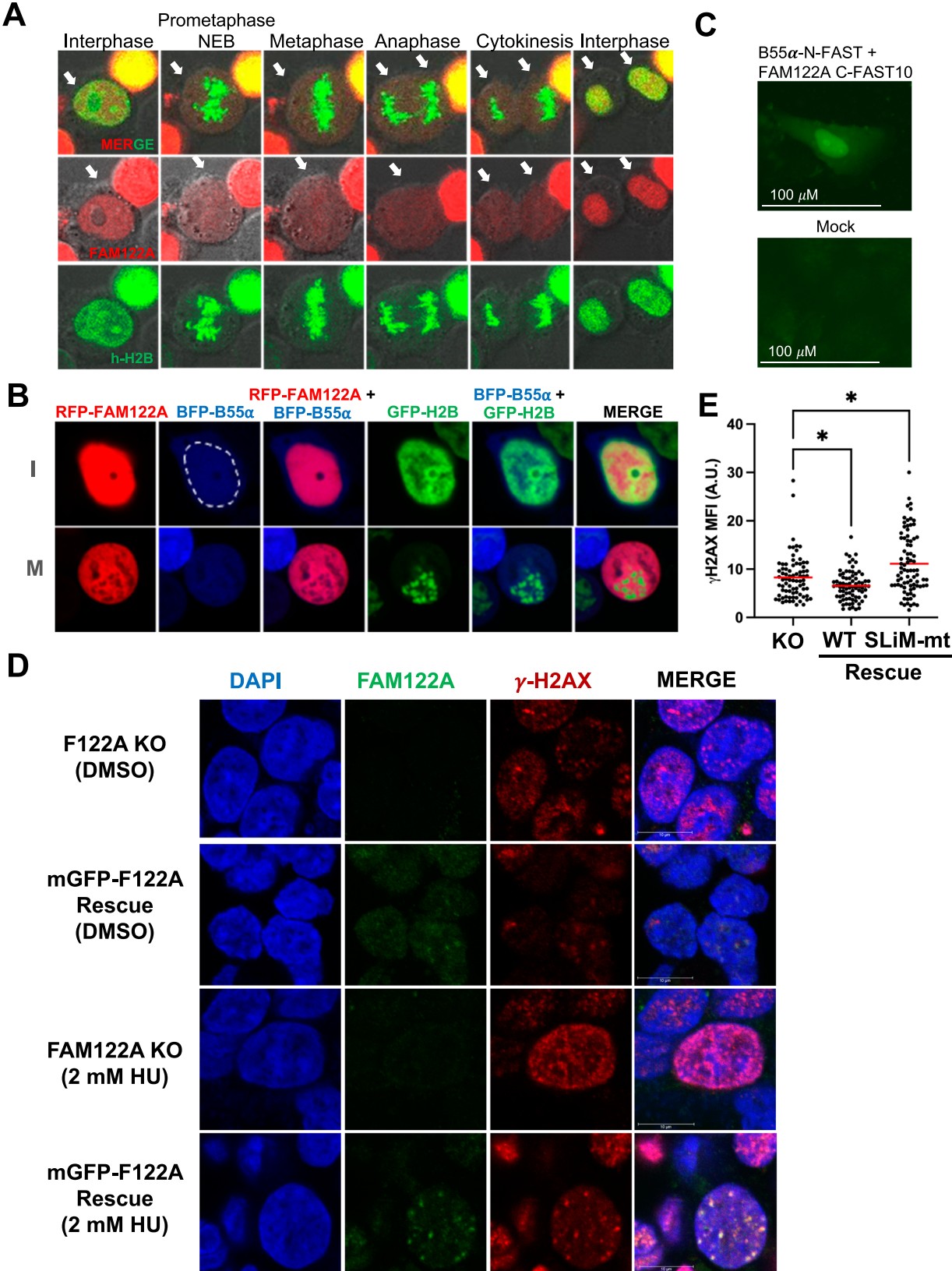

demonstrating that FAM122A binding to B55α is essential for checkpoint activation. In the absence of FAM122A, cyclin E expression is reduced, and CHK2 and CHK1 activities are attenuated in response to additional replication stress. We also observed slower progression through the G0/G1 transition, resulting from attenuation of mitogenic signaling, diminishing the accumulation of cyclin D1 and pRB

phosphorylation. Surprisingly, FAM122A interaction with B55α is largely constant during the cell cycle and upon mitotic checkpoint activation. Regulation may be restricted to modulation of FAM122A inhibitory function potentially through changes in FAM122A interactions regulated by posttranslational modifications within the IDRs. This was unexpected given recent work describing that CHK1

**Fig. 9 | FAM122A is localized in the nucleus in interphase and to DNA damage foci in response to replication stress. A** mRFP-FAM122A wild-type was co-transfected with GFP-h-H2B into HEK293T cells and observed by time-lapse confocal microscopy (20x objective, 2.5x zoom factor). Arrows indicate the cell(s) in the cell cycle phase. **B** mRFP-FAM122A, BFP-B55α and GFP-h-H2B expressing vectors were transfected into HEK293T cells to monitor localization. Close up of two cells in interphase (I) and in Mitosis (M) are shown (the nuclear envelope is traced with a dashed white line to demonstrate expression of B55α in both the cytoplasm and the nucleus). **C** U-2 OS cells were transfected B55α and FAM122A SplitFast constructs and colocalization was visualized by adding the HMBR fluorogen to the media 48 h later. Experiment performed as biological triplicate. **D** FAM122A KO and

reconstituted WT expressing C1 HEK293 cells were treated with 2 mM HU for 24 h, stripped of cytoplasmic proteins and fixed for immunofluorescence imaging to observe localization of FAM122A bound to chromatin in the context of γH2AX foci on a confocal microscope using a 63x objective and 5x zoom factor. The colocalization of FAM122A with γH2AX foci are represented by the yellow signal. Experiment performed as biological triplicate. **E** FAM122A KO and FAM122A reconstituted HEK293 cells (WT and SLiM MT) were treated with 2 mM HU for 3 h and subjected to gamma γH2AX immunofluorescence where mean-fluorescent intensity was analyzed using a Kruskal-Wallis test (two-sided) with a Dunn's multiple comparison test. KO:WT Recon. p-value = 0.0429. KO:SLiM MT Recon. p-value = 0.0141. n = 3 biological replicates. Source data are provided as a Source Data file.

mediated phosphorylation of FAM122A results in its sequestration by 14-3-3 proteins and activation of PP2A/B55α that leads to WEE1 dephosphorylation and stabilization[23]. We have not observed 14-3-3 binding to FAM122A. Instead, we observe that replication stress induces formation of FAM122A foci at DNA damage sites marked by γH2AX. These data are also consistent with DepMap data that show an inverse correlation in the effect of ablation of *PPP2R2A* with respect to *CHEK2 and TP53* in human cell lines (Supplementary Fig. 8H)[40].

## Methods

### Cell culture and cell lines
All cell lines were obtained from ATCC and cultured in DMEM supplemented with 10% TET-Free FBS and 0.1% Penicillin-Streptomycin as described previously[18] and tested for mycoplasma biannually or if there were any signs of stress. For lentivirus generation, HEK293T cells were transfected with pCMV-VSVG, pCMV-deltaR8.2, and expression constructs. Lentiviral supernatants were concentrated up to 100-fold using Takara Bio Lenti-X Concentrator. To overexpress FLAG-B55α in HEK293 cells, cells were transduced twice with pCW57.1 FLAG-B55α 24 h from one another and selected with puromycin and clonally isolated to determine expression levels among clones post doxycycline expression (48 h). To genetically ablate *FAM122A*, HEK293, T98G, and U-2 OS cells were transduced with one of two lentiCRISPRv2-*sgFAM122A* (#1/#2) and selected with puromycin for clonal isolation. For Dox-inducible mRFP-FAM122A WT/Mutant cell line generation, T98G *sgFAM122A* #2 KO cells were transduced with pCW57.1-mRFP-*Fam122a* WT/MUT lentiviruses and bulk sorted according to mRFP fluorescence following doxycycline induction (48 h). For mGFP-FAM122A WT/SLiM MUT HEK293 rescue cells, cells were transfected with pCMV6-AN-mGFP-*Fam122a* WT/SLiM MUT constructs with Lipofectamine 3000 and selected with G418 (500 μg/ml) followed by clonal isolation.

### Plasmids
Plasmids used or generated in this study are described in the Key Resources Table (Supplementary Data 6). pCMV6-*Fam122a*-Myc-DDK was purchased from Origene. pCMV6-AN-DDK-*Fam122a* was generated using SgfI/MluI restriction sites from the precision shuttle system (pCMV6-AN-DDK-Pol Iota- A kind gift from Roger Woodgate Addgene #131228). pCMV6-AN-mGFP and pCMV6-AN-mRFP plasmids were kind gifts from Richard Katz with Fam122a inserted using SgfI/MluI restriction sites. pGEX2T-*Fam122a*, pCMV6-AN-DDK-*Fam122a*, pCMV6-AN-mGFP-*Fam122a*, pCMV6-AN-RFP-*Fam122a*, pCW57.1-mRFP-*Fam122a*, and pGEX2T-FRs variant mutants were generated by site-directed mutagenesis using QuikChange II XL Site-directed mutagenesis (Agilent) with primers listed in the Key Resources Table (Supplementary Data 6) and subsequently validated by Sanger sequencing. pGEX2T-FR deletion constructs were generated by PCR cloning BamHI/EcoRI restrictions sites, digestion, and ligation to pGEX2T. pCW57.1 FLAG-B55α and pCW57.1-mRFP-*Fam122a* were generated by LR clonase reactions (ThermoFisher Scientific) from the ORFs cloned in pENTR1A. (BamHI/EcoRI for mRFP-*Fam122a* and BglII/EcoRI for FLAG-B55α).

### Transfections
Transient transfections of Myc- and FLAG-containing constructs was achieved using calcium phosphate transfection or Lipofectamine 3000. In the case of calcium phosphate, 5 μg plasmid DNA was added dropwise to 2× HEPES-buffered saline (HBS) solution (280 mM NaCl, 50 mM HEPES, 1.5 mM Na2HPO4, pH 7.05) with bubbling, followed by addition to cells treated with 25 mM chloroquine after a 30 min incubation period. Alternatively, the transfection of stable cell lines was performed using Lipofectamine 3000 according to manufacturer's recommendations.

### siRNA transfection
siRNA for human *FAM122A* and *NC1* (negative control 1) were purchased from IDT. 10 nM final concentration of *siFAM122A* was transfected in T98G cells between 50–60% confluency using Lipofectamine RNAiMax and incubating for 48 h in serum-free conditions.

### Sample processing protocol
HEK293 iFLAG-B55α clones 5 (high expresser) and 6 (lower expresser) were induced with 1 μg/mL of doxycycline for 24 h. Two mg protein was precipitated and digested with 1% Protease Max (Promega) and 20 μg of Promega Trypsin/LysC mix for 4 hours at 37 °C with vigorous shaking. Samples were reduced with 5 mM DTT and alkylated with 10 mM iodoacetamide at room temperature. Samples were then digested with sequencing-grade modified trypsin (Promega) overnight at 37 °C in an incubator shaker. Peptides were cleaned by subsequent elution from C-18 and Hypercarb/Hypersep PGC columns (Thermo Scientific) and dried via speed-vac. TiO2 beads (GL Sciences) were resuspended in binding buffer (2 M lactic acid in 50% ACN) at a concentration of 100 μg/μl and added to the peptides at a ratio of 4 mg beads per 1 mg peptide. Samples were vortexed and incubated 2 × 30 min with end over end rotation. The beads from both incubations were combined and washed three times with 1 mL binding buffer, then three times with 1 mL 50% ACN. Peptides were eluted from the TiO2 beads with 600 μL 5% ammonium hydroxide in 50% ACN by vortexing and passed through C-18 stage tips. Eluted peptides were dried in a speed-vac and subsequent LC-/MS/MS analysis was performed. Proteolytic peptides were resuspended in 0.1% formic acid and separated with a Thermo Scientific RSLC Ultimate 3000 on a Thermo Scientific Easy-Spray C18 PepMap 75μm × 50 cm C-18 2 μm column with a 240 min gradient of 4–25% acetonitrile with 0.1% formic acid at 300 nL/min at 50 °C. Eluted peptides were analyzed by a Thermo Scientific Q Exactive plus mass spectrometer utilizing a top 15 methodology in which the 15 most intense peptide precursor ions were subjected to fragmentation. The AGC for MS1 was set to $3 \times 10^6$ with a max injection time of 120 ms, the AGC for MS2 ions was set to $1 \times 10^5$ with a max injection time of 150 ms, and the dynamic exclusion was set to 90 s.

### Phosphoproteomic mass spec analysis
HEK293 iFLAG-B55α clones 5 (high expresser) and 6 (lower expresser) were induced with 1 μg/mL of doxycycline for 24 h. To identify potential substrates and downstream targets of B55α in +/−Dox

treated HEK293-iFLAG-B55α cells, global phosphoproteomics analysis was performed as previously described[41–43] Raw data analysis was performed using MaxQuant software 1.6.1.0 and searched against the Swiss-Prot human protein database (downloaded on July 26, 2017, 20214 entries). The search was set up for full tryptic peptides with a maximum of two missed cleavage sites. All settings were default and searched using acetylation of protein N terminus, oxidized methionine, and phosphorylation of Ser, Thr and Tyr being included as variable modifications, with a maximum number of modifications per peptide being 5. Carbamidomethylation of cysteine was set as fixed modification. The precursor mass tolerance threshold was set at 10 ppm and maximum fragment mass error was 0.02 Da. LFQ quantitation was performed using MaxQuant with the following parameters; LFQ minimum ratio count: 2, Fast LFQ: selected, LFQ minimum number of neighbors; 3, LFQ average number of neighbors: 6. Global parameters for protein quantitation were as follows: label minimum ratio count: 1, peptides used for quantitation: unique, only use modified proteins selected and with normalized average ratio estimation selected. Match between runs was employed and the significance threshold of the ion score was calculated based on a false discovery rate of <1%. The MaxQuant normalized ratios "Phospho(STY).txt" were used for quantitation data analysis in Perseus software (1.6.5)[44]. Phosphorylated serine, threonine, and tyrosine (pSTY) sites were filtered for only those that were confidently localized (class I, localization probability ≥ 0.75) followed by filtering for phosphosites identified in at least 70% of runs. Quantitative differences amongst treatments were determined using a two-sample Student's t-test with the following parameters, (S0 0.1, and Side, Both) using p value < 0.05. All detected phosphosites (regardless of p value) were used to cross-reference the motif in ScanProsite.

## Affinity-purification mass spec analysis

Proteins from affinity-purified pull-downs were precipitated using either 20% tri-chloroacetic acid (TCA), washed twice with acetone (Burdick & Jackson, Muskegon, MI) or the single-pot, solid-phase enhanced method[45]. Proteins were digested overnight in 25 mM ammonium bicarbonate with trypsin (Promega) for mass spectrometric analysis. Peptides were analyzed on a Q-Exactive Plus quadrupole equipped with Easy-nLC 1000 (ThermoScientific) and nanospray source (ThermoScientific). Peptides were resuspended in 5% methanol/1% formic acid and analyzed as previously described. Raw data were searched using COMET (release version 2014.01) in high resolution mode[46] against a target-decoy (reversed)[47] version of the human proteome sequence database (UniProt; downloaded 2/2020, 40704 entries of forward and reverse protein sequences) with a precursor mass tolerance of +/− 1 Da and a fragment ion mass tolerance of 0.02 Da, and requiring fully tryptic peptides (K, R; not preceding P) with up to three mis-cleavages. Static modifications included carbamidomethylcysteine and variable modifications included oxidized methionine. Searches were filtered using orthogonal measures including mass measurement accuracy (+/− 3 ppm), Xcorr for charges from +2 through +4, and dCn targeting a <1% FDR at the peptide level. Quantification of LC-MS/MS spectra was performed using MassChroQ (Valot et al., 2011) and the iBAQ method[48]. A total peptide count > 1 at least two quantifications in either the WT or MT sample were required for inclusion in the dataset. Proteins with a total peptide count >1 were removed as non-specific binders. Missing values were imputed, and statistical analysis was performed in Perseus[44].

## Cell cycle, flow cytometry, and cell treatments

Parental and KO T98G cells were serum starved for 3 days prior to mitogenic stimulation with serum-containing media and collected at the indicated timepoints for flow cytometry and WB analysis. Collection for flow cytometric DNA content analysis was performed by collecting the cells followed by ice-cold ethanol fixation. Following 2 PBS washes, cells were stained with 1x propidium iodide solution (in 1% FBS in PBS with 1 mg/mL RNase A) in the dark for 30 minutes.

For cells expressing mGFP (TagGFP), cells were fixed in ice-cold methanol to preserve GFP fluorescence and processed as samples fixed with 70% ethanol.

BrdU incorporation assays in HEK293 cells were performed by the addition of 10 μM BrdU to cells for a 45-minute pulse. Cells were fixed using ice-cold 70% ethanol and left overnight for permeabilization. Cells were washed with PBS twice followed by denaturing with 2 N HCl for 20 min. Following washes and neutralization with 0.1 M $Na_2B_4O_7$ for 10 min, cells were incubated with anti-BrdU antibody (1:200) for 20 min. Subsequent washes were followed up by anti-rabbit Alexa Fluor 488 secondary (1:200) for 10 min. Following two additional washes, cells were stained with 1x propidium iodide solution as described above.

EdU incorporation assays were performed using the APEXBio EdU Flow Cytometry Assay Kits (Cy5) (K1078). Briefly, a 10 μM final concentration of EdU was added to cells 1 h prior to collection and the cells washed and fixed in 4% PFA in PBS. Cells were permeabilized using a 1x saponin-based permeabilization agent (PBS pH 7.4), 1% BSA, 0.1% NaN3, and 0.1% saponin) and subject to the Cy5 azide click-chemistry reaction. Following additional wash steps with the saponin-permeabilization reagent, cells were stained with 1x propidium iodide solution.

All flow cytometry experiments were carried out on a BD LSR-II and analyzed in FlowJo v10 (BD Biosciences). For nocodazole release, cells were initially treated with 10 nM nocodazole for 20–24 h followed by washout and collections at indicated timepoints. For RO3306 (CDK1i), cells were treated with 10 μM for 24 h and either collected or released following PBS washout and collected at given time points for PI staining as described above. For Thymidine and HU Challenge/Release experiments cells were treated with 2 mM Thymidine or HU for given time points (24 h in the case of release) and collected or released with collections at the given timepoints post-release. For CHK1i (pre-xasertib) treatments cells were treated with 100 nM for 24 h and collected.

## Clonogenic growth and proliferation analysis

Proliferation curves were established by seeding 1000, 2000, 3000, and 4000 cells per well with 4 wells per condition and the area phase contrast measured every 4 hours with an IncuCyte SX5. For clonogenic assays, cells were cultured for 11 days followed by fixation and stained with crystal violet.

## GST pulldown assays

GST-tagged constructs of interest were expressed in Escherichia coli bacteria and purified for use in pull-down assays. Briefly, 100 mL cultures of E. coli were treated with 0.25 mM isopropyl β-D-thiogalactoside (IPTG) for 3 h to induce expression of GST-fusion proteins. Cells were then harvested by centrifugation and resuspended in NETN lysis buffer (20 mM Tris pH 8, 100 mM NaCl, 1 mM EDTA, 0.5% NP-40, 1 mM PMSF, 10 μg/mL leupeptin) prior to sonication at 30% amplitude for 10 cycles. Supernatants were collected and incubated with glutathione beads for purification, followed by NETN buffer washes for sample clean-up. For pull-down assays, purified GST-FAM122A, deletion, and mutant constructs were incubated with HEK293T lysates for 3 hr or overnight at 4 °C, followed by washes (4×) with complete DIP lysis buffer (50 mM HEPES pH 7.2, 150 mM NaCl, 1 mM EDTA, 2.5 mM EGTA, 10% glycerol, 0.1% Tween-20, 1 μg/mL aprotinin, 1 μg/mL leupeptin, 1 μg/mL Pepstatin A, 1 mM DTT, 0.5 mM PMSF) and elution with 2× LSB. Samples were resolved by SDS-PAGE and probed using antibodies against proteins of interest.

## Determining FAM122A and p107 R1-R2 affinity of B55α

To determine the affinity of the FAM122A:B55α and p107-R1-R2:B55α interactions we used the affinity by depletion method described previously[30]. Briefly, the concentrations of GST-fusion proteins were determined by SDS-PAGE with the use of a BSA standard and Coomassie blue staining. Selected concentrations of GST-fusion proteins were determined by their calculated molecular mass from the amino acid sequence and were incubated with 100 μg of HEK293T lysate in a 100 μl volume overnight to achieve equilibrium. The pulldowns were centrifuged at 4000 × g and the supernatants collected without disruption of the GST-fusion protein bead pellet. Supernatant samples (30 μg) were resolved by SDS-PAGE and probed for B55α and GAPDH using the antibodies listed. The resulting WBs were quantitated using Fiji ImageJ by selecting the B55α and GAPDH bands while subtracting the background for each membrane and normalizing the B55α signal to the GAPDH intensity. Lastly, these measurements were compared against the supernatant without GST-Fusion protein, and the inverse measurement was calculated to determine the fraction of B55α bound to the GST-fusion proteins.

## In vitro kinase and phosphatase assays

GST-tagged p107 constructs loaded on beads were phosphorylated with recombinant cyclin B/CDK1 (Invitrogen) in 200 nM ATP KAS buffer (5 mM HEPES pH 7.2, 1 mM MgCl2, 0.5 mM MnCl2, 0.1 mM DTT) as described previously[49]. We incubated samples with shaking at 37 °C for 2 h unless indicated. Reactions were stopped by adding 2× LSB and heated at 65 °C when used for PAGE or immunoblots. In vitro phosphorylated substrates for phosphatase assays were washed 3× in complete DIP buffer, followed by addition of indicated concentrations of purified PP2A/B55α[49]. Reactions were incubated with shaking at 37 °C for times indicated, followed by addition of 2× LSB and boiling at 65 °C for western blotting.

DiFMUP phosphosubstrate reactions were performed using the Invitrogen EnzChek™ Phosphatase Assay Kit (E12020). Purified FLAG-B55α/PP2A holoenzymes were preincubated with soluble HA-FAM122A WT or MUT for 15 min at RT. Upon the addition of 200 μM DiFMUP in a 1:1 volume, the reactions were shielded from light, and the substrate fluorescence measured at 15 min intervals at 365 nm wavelength.

## Eluted Fam122a for competition assays

A PreScission protease site and 2x-HA tag were added to pGEX2T-Fam122a via restriction enzyme cloning. GST-protein purification was performed as described above. 2x-HA-Fam122a was eluted off GSH beads by PreScission protease cleavage in PreScission protease cleavage buffer (50 mM Tris-HCl, pH 7.0 (at 25 °C), 150 mM NaCl, 1 mM EDTA, 1 mM dithiothreitol) at 4 °C overnight. Following brief centrifugation, the eluate was collected and prepared to contain 25% glycerol and 1 mM DTT for −80 °C storage. Briefly, purified PP2A/B55α holoenzymes were preincubated with increasing nM concentrations of Fam122a for 30 min at 37 °C to facilitate interaction. These preincubated PP2A/B55α complexes were then incubated with μM concentration GST-tagged p107 R1R2 constructs for 3 h or overnight at 4 °C, followed by washes (4x) with DIP lysis buffer and elution with 2× LSB. Proteins were resolved via SDS-PAGE and detected via western blotting using anti-B55α and GST antibodies.

## Immunoprecipitations

Whole-cell extracts (200–1000 μg) were incubated with Myc- or FLAG-conjugated beads (Sigma) for 3 h or overnight at 4 °C. Input samples were collected prior to antibody-conjugated bead incubation, and supernatants were taken post-incubation following sample centrifugation. Beads were then washed (4×) with complete DIP lysis buffer and proteins were eluted in 2× LSB. Proteins were resolved by SDS-PAGE and probed using antibodies against proteins of interest.

## DNA fiber assay and analysis

HEK293 Parental and FAM122A KO cells were grown to 60–70% confluency and treated with 50 uM CldU for 30 min followed by a PBS washout and 250 uM IdU for 90 min. After collection, PBS washes and resuspension, cells were lysed directly on slides (200 mM Tris-HCl, 50 mM EDTA, 0.5% SDS, pH 7.4). The slides were tilted and left to air dry to generate DNA spreads. The DNA spreads were fixed in a Methanol: Acetic acid mixture (3:1) and left to dry. To expose the epitopes immunofluorescence, the DNA spreads were denatured in 2.5 M HCl, washed, subject to BSA blocking, and incubated in primary antibodies (Mouse anti-BrdU (IdU) (BD Biosciences 347580) (1:100) and Rat anti-BrdU (CldU) (Abcam ab6326) (1:300). Following PBST washes, the spreads were stained with secondary antibodies Anti-mouse Alexa594 (Life Technologies A11032) Anti-rat Alexa488 (Life Technologies A110006) both at 1:300, washed and mounted in Prolong diamond mounting media.

Fibers were imaged on a Nikon Eclipse Ni epi-fluorescent microscope using an oil-immersion 40x objective. Images were acquired using the NIS-Elements software. All samples were blinded when being imaged and analyzed. At least 100 fibers were scored per experiment, which were repeated in triplicate. The data shown were pooled from independent experiments. Only continuous fibers with clearly defined beginnings and ends of each color were scored. Fiber length was quantified manually using the freehand line tool in Fiji ImageJ (NIH). Both CldU and IdU fiber lengths were quantified. Pixel values were converted to microns using the scale generated by the NIS-Elements software during imaging. The values were then plotted, and the graphs generated using Prism GraphPad 9 software.

## Protein analyses

Cells were lysed with ice-cold lysis buffer (50 mM Tris-HCl (pH 7.4), 5 mM EDTA, 250 mM NaCl, 50 mM NaF, 0.1% Triton X-100, 0.1 mM $Na_3VO_4$, 2 mM PMSF, 10 μg/ml leupeptin, 4 μg/ml aprotinin, and 40 μg/ml Pepstatin A) or Complete DIP. Proteins were resolved by 8% or 10% polyacrylamide/SDS gel electrophoresis and transferred to a polyvinylidene difluoride (PVDF) membrane (Immobilon-P, Millipore) in 10 mM CAPS/10% methanol buffer (pH 11). Bands were visualized by using SuperSignal West Pico PLUS Chemiluminescent Substrate (Thermoscientific) and imaged using an iBright CL1000 imaging system (ThermoFisher). Antibodies used are listed in the Key Resource Table (Supplementary Data 6). Densitometric analysis of western blots was performed in Fiji (ImageJ)[50].

## Graphs and statistical analysis

All experiments were performed in triplicate unless otherwise specified. Graphs depict calculated SEM from all replicates. Graphs were generated in Prism GraphPad 9 and 10. Statistical analyses for densitometric analyses were performed using one-way ANOVA with Dunnett's correction for multiple comparisons. Statistical analysis for DNA fiber length assays was determined using Mann-Whitney U test. The analysis of mean fluorescent intensities of immunofluorescence staining was performed using Kruskal-Wallis tests with Dunn's corrections following the determination of dataset normality. Statistical analyses of proliferation curves comparing parental to FAM122A knockout cell lines was conducted using a Wilcoxon matched-pairs signed rank test. In the case of WT FAM122A and SLiM MUT rescues, a Friedman test was performed of the last 10 measurements comparing all conditions to FAM122A KO cells. Statistical significance of $p$-values is represented as follows: *<0.05, **<0.01, ***<0.001, and ****<0.0001.

## Immunofluorescence staining and fluorescent microscopy

Cells were seeded on iBidi 8 well glass bottom (Cat. 80827) μ-slide treated previously with Poly-D-Lysine (Cat. A389040). Cells were treated with hydroxyurea (2 mM for 24 h or 4 mM for 6 h), 100 nM Prexasertib for 24 h. Cells were treated as previously described[51] for

whole cell staining or as described for chromatin bound staining[36] using α-FAM122A (1:50), α-γH2AX (1:200), and α-B55α (2G9) (1:50) antibodies. Anti-rabbit AlexaFluor 488 and Anti-mouse AlexaFluor 594 secondary antibodies were used at 1:100 dilution. All confocal microscopy was performed on a Leica TCS SP8 microscope using 63x oil immersion lens. Other imaging was performed using an EVOS cell imaging system.

## AlphaFold2 calculations
Some calculations were performed using the Jupyter notebooks provided by ColabFold[28], which requires inputting the query sequence and choosing options for running the calculation using Google Colab GPUs. We used both the original AlphaFold2 notebook and the Alpha-Fold2_advanced notebook (https://colab.research.google.com/github/sokrypton/ColabFold/blob/main/beta/AlphaFold2_advanced.ipynb). The latter was developed to model complexes of proteins before the availability of AlphaFold-Multimer. The notebook https://colab.research.google.com/github/sokrypton/ColabFold/blob/main/AlphaFold2.ipynb implements both the original AlphaFold2 trained on single-chain proteins and AlphaFold-Multimer for complexes. We ran AlphaFold2 without templates on the ColabFold notebooks.

For calculations with the recently released AlphaFold-Multimer 2.3 (December 2022), we downloaded code from the DeepMind github repository (https://github.com/deepmind/alphafold) for computations on a Linux workstation with a 24 Gbyte GPU. All version of AlphaFold2 run five separate models consisting of the weight matrices that convert input features (such as the query sequence and the multiple sequence alignment of homologues) into predicted structures. The first two models use templates. Models 3, 4, and 5 do not. Structure predictions for the tetrameric complex of PP2A/A, B55α, PP2A/C, and FAM122A were performed with the use of templates (PDB70) in models 1 and 2. Each set of 5 models from AlphaFold-Multimer was run with five random seeds for a total of 25 models. Models were ranked as recommended in the AlphaFold-Multimer paper[29], which is a linear combination of the pTM and iPTM scores (0.8*iPTM + 0.2*pTM), which combines the "interaction" predicted TM score (iPTM) and the whole structure predicted TM score (pTM). The top model of the tetrameric complex had an iPTM +pTM score of 0.804. This structure prediction came from AlphaFold-Multimer model4, and so did not use experimental structures as templates. Structures were optimized with AMBER (default in AlphaFold2) and visualized with PyMOL. The model is available in Supplementary Data 4 and its coordinates in Supplementary Data 5.

## Reporting summary
Further information on research design is available in the Nature Portfolio Reporting Summary linked to this article.

## Data availability
The AlphaFold-Multimer model of the tetrameric complex and a PyMOL session data generated in this study have been deposited in the Zenodo under accession code DOI 10.5281/zenodo.7739038: https://doi.org/10.5281/zenodo.7739038.The mass spec data generated in this study for Fig. 4C have been deposited in the ProteomeXchange database under accession code *PXD052731*: MassIVE: https://massive.ucsd.edu/ProteoSAFe/dataset.jsp?task=7216b37288104812881bb97607f0fef7. The mass spectrometry proteomics data generated in this study for Fig. 1B have been deposited to the ProteomeXchange Consortium via the PRIDE partner repository with the dataset identifier PXD052836 Source data are provided with this paper.

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

## Acknowledgements

This work was supported in part by the National Institutes of Health Grants R01 GM117437 and R03 CA216134-01, a WW Smith Charitable Trust Award, R35 GM122517 (R.L.D.), R01 CA211670, R01 CA282766 (J.S.D.), R35 GM119455 (A.N.K.), a Pre-Pilot Award from U54 CA221704 (J.S.W., Z.Z., and H.F.), T32GM142606 Training fellowship to (D.J.G.) and funding from NCI CCSG grant P30 CA006927 (X.G., J.D., N.J. and R.L.D.). We also thank the Molecular Modeling Facility at the FCCC.

## Author contributions

X.G. and J.S.W. wrote the first draft of the manuscript. R.L.D., J.S.D., A.N.K., Z.Z. and N.J. edited the manuscript. X.G., R.L.D. and J.S.W. designed experiments. N.J., J.S.D., and A.N.K., contributed to the experimental design. B.F., K.R.P., A.M.K., S.M.P., D.J.G. H.F., Z.Z. performed specific experiments contributing to the main and or Supplementary Figs. B.F., Q.X. and R.L.D. performed data-guided computational model and docking predictions. X.G. and J.S.W. performed additional analysis of the models and docking, B.C.M., L.C., A.N.K., J.S.W., A.M.K and J.S.D. performed mass spec analysis.

## Competing interests

The authors declare no competing interests.
