## [Peer Review File · Nature Communications]

FAM122A ensures cell cycle interphase progression and checkpoint control by inhibiting B55/PP2A through helical motifsReviewers' Comments:

Reviewer #1:

Remarks to the Author:

Although FAM122A has been proposed as a PP2A/B55 α holoenzyme inhibitor previously, there is really a lack of the evidence for the structural analysis of FAM122A interacting or inhibiting PP2A/B55 α associated with its substrates. In this study, the authors studied the main interaction regions of FAM122A with PP2A/B55 α by computational structure prediction and mutational and biochemical methods, and showed that conserved SLiM is necessary for FAM122A binding to B55 α in vitro and in cells, and also showed the spatial structure prediction of how FAM122A interrupts the interaction of B55 α and its substrate P107. These data are novel for this field and essential for the understanding the biology of FAM122A. Overall, the study is well-designed and supported by most experimental data.

Major comments:

1. Whether the SLiM is critical for the inhibitory effect of FAM122A KO on cell growth is not very convince. In Fig. 6, the rescue of the growth inhibitory effect in FAM122A KO should be compared in the FAM122A KO cells re-expressed by FAM122A WT and FAM122A SLiM mutant cells.
2. Also, the reviewer wondered if there is any try for the crystal structure experiments of FAM122A interacting with PP2A/B55 α with or without substrate.
3. In Fig.9D, FAM122A signaling is too weak and invisible. Is there any difference for DNA damage response between FAM122A KO and mGFP FAM122A upon HU treatment? Here just show one field and not clear the total effects.
4. Is the SLiM is necessary for the protection of FAM122A-expressing cells challenged by HU treatment. Or giving the discussion for any other possibility?
5. There are several missing data in manuscript. Such as missing Suppl. Fig.3D, no Suppl. Fig. 4?

Some minor points:

1. What's the meaning of the symbols, such as ".", ":" and "*" in Fig. 1.
2. In Fig.3, the SLiM "RLHQIK" is not the same as stated in manuscript.
3. In Fig.7D, the annotations of pGSK and pERK in the right of the panels should be properly indicated.
4. In all western blot, the molecular weights should be marked.

Reviewer #2:

Remarks to the Author:

The manuscript by Wasserman et al, "Fam122A ensures cell cycle interphase progression and checkpoint control as a SLiM-dependent substrate-competitive inhibitor to the B55alpha/PP2A phosphatase, presents a lot of interesting data and modeling concerning the relationship of FAM122A to B55alpha/PP2A. The first part of the paper investigates how FAM122A binds to the B55alpha subunit and convincingly demonstrates that this interaction is mediated by a conserved 'SLiM', which actually seems to form an alpha helix (so is it actually a SLiM or more broadly defined as a docking motif?). While there is no structure presented, there is extensive modeling presented with different versions of alpha-fold and extensive mutagenesis and interaction assays (in vivo, in vitro, competition with p107 which contains the same docking motif) etc. This is the strongest part of the paper, and demonstration of key residues in B55alpha that are required for the interaction w/ FAM122A together with mutants of FAM122A make a very strong case for this 'SLiM'-dependent interaction. The second part of the model proposes that a longer helix C-terminal to the B55alpha docking helix forms a direct interaction with the active site of the PP2AC catalytic subunit. This part of the model is much less convincing as two mutants (conserved E residues changed to K) have no effect on observed FAM122A-B55alpha/PP2A interactions (through qualitative pull downs) and were not tested for effects on in vitro dephosphorylation. However, this paper successfully challenges an earlier study that proposed FAM122A inhibited B55alpha/PP2A by promoting degradation of the catalytic subunit.

Finally, the last part of the paper analyses in vivo effects of FAM122A deletion/reexpression with SLiM mutants. Effects on proliferation are clear (although were demonstrated in the 2016 paper). More interesting are demonstrated effects of FAM122A on cell cycle entry (via regulation of p107/RB protein phosphorylation) and regulation of G1/S checkpoint control during replicative stress. However, localization analyses proposed to demonstrate that FAM122A localizes exclusively to the nucleus during interphase, are not (in current form) convincing.

Overall, the paper has important findings but the overall impact is reduced by significant flaws. The paper is quite long and could benefit from eliminating some redundancies to focus on the key experiments that provide the most informative and novel results and indication of mechanisms for observed effects in vivo. See below for specific comments and suggestions for modifications to strengthen the study.

1. In Figure 1 authors present their pipeline for identifying proteins containing the proposed B55alpha binding 'SLiM' which combines bioinformatic analyses with cross validation with several published experimental data sets. While it is clear that FAM122A is the 'winner' from these analyses some discussion of other candidates that came out from these extensive analyses should also be included. In particular, candidates that in figure 1B were identified in 4/5 of 5/5 data sets should be listed along with their proposed SLiMs and identifying which were previously identified (i.e. p107 etc). Currently supplemental table 1 only contains information about phosphoproteomic analyses carried out by these authors.

2. Mutagenesis of the 'SLiM' and complementary mutagenesis of B55 alpha supports the predicted model strongly. Authors state, however that S73A mutation abolishing a phosphosite amino-terminal to the SLiM had no effect on interaction. Confidence in this statement is reduced by the fact that no quantitative assays are performed to determine affinity of the wt FAM122A or mutants for B55alpha/PP2A, and there is no evidence presented or discussed concerning whether this site is ever phosphorylated, and if the wild-type FAM122A would have been phosphorylated at this site under co-IP conditions used to test effects of the S73A mutant. In general, the lack of quantitative affinity determinations are a weakness in the paper.

3. In Figure 2 analysis of conserved regions is presented. FR4 contains BOTH predicted helices as shown in the next section of the paper and this should be emphasized. There is no direct evidence that the second helix forms or that it interacts w/ catalytic active site of PP2AC, which is the obvious missing link in the authors' proposed mechanism. Besides modeling, there is no convincing support presented. Analyses of FAM122A binding to B55alpha WT vs mutant C239G are consistent with FAM122A binding to the holoenzyme through additional interactions, but this is an indirect argument and does not identify where the additional interactions may be. Furthermore a potentially interesting effect on FAM122A phosphorylation is pointed out (Figure 3E), but not explained or discussed further. Rather, direct experiments that examine any effects of mutations proposed to disrupt this second helix on FAM122A-B55alpha/PP2A binding and/or dephosphorylation should be included. Most important, authors propose that E97 and E101 contact Mn²⁺ residues in the catalytic active site and that changing these to K does not affect qualitative assays of FAM122A-B55alpha PP2A binding. To identify any ability of FAM122A to inhibit the catalytic center of PP2AC, authors should examine in vitro dephosphorylation of a small molecule (pNPP or DiFMUP) that does not depend on SLiM-dependent docking, by the B55alpha heterotrimer +/- FAM122A wt or E97,101K mutants.

4. Discussion of different versions of AlphaFold structural predictions make up a large portion of the paper. Since the newest version of AlphaFold Multimer v2.3 seems to provide the most comprehensive information, I suggest editing the paper to contain a single section where the most confident structural prediction is presented and discussed.

5. The next part of the paper analyses in vivo effects of FAM122A on proliferation/colony formation (Fig6), and shows it is SLiM dependent. This strongly supports the structural model and demonstration that p107 and FAM122A compete for binding to B55alpha/PP2A holoenzyme.

6. In Fig.s 7 , 8 and 9, authors investigate cell cycle entry/progression, as well as effects on G1/S checkpoint activation in cells lacking FAM112A. These are complicated in vivo assays, and would be

easier for a non-expert to appreciate by highlighting what the direct B55alpha/PP2A substrates are in each case, and if deletion of FAM122A causes the observed biological effects through increased activity of the phosphatase as one would predict from deletion of an inhibitor. As is, the studies seem mainly descriptive.

7. Figure 9 presents potentially interesting analyses of FAM122A localization, and identifies an NLS. However, in Figure 9A images, it is very difficult to see RFP-FAM122A signal and similarly in figure 9D, FAM122A and gammaH2Ax signals are barely detectable and would be better appreciated perhaps if shown as high contrast black/white images. Finally, no information is presented about any phenotypes observed when the NLS is mutated, so it is difficult for the reader to appreciate the potential biological relevance of the reported nuclear localization of FAM122A.

Reviewer #3:

Remarks to the Author:

In the manuscript entitled "FAM122A ensures cell cycle interphase progression and checkpoint control as a SLiM-dependent substrate-competitive inhibitor to the B55 α /PP2A phosphatase", the authors explore the biological functions of FAM122A protein, a proposed inhibitor of the phosphatase PP2A/B55a holoenzyme. The manuscript is separated in two parts, the first one being mostly dedicated to identify amino-acid residues involved in FM122A - PP2A/B55a interaction and the second one investigating cell cycle defects upon FAM122A knock-out.

Overall, I found this work interesting, but I have several concerns about the main conclusions, as detailed below.

Also, I found this manuscript difficult to read as figure legends frequently do not describe/explain the different conditions used.

Main points:

. The authors propose that FAM122A acts as a competitive inhibitor and exhibits a higher affinity for PP2A/B55a than other substrates, such as RB-related protein p107.

First, it is unclear in Figure 2E if the authors compared (by GST pulldown) the affinity for B55a of a full-length FAM122A (FL) protein versus a truncated p107 construct (p107 R1-R2). Second, to conclude on in vitro competition assays using purified HA-FAM122A versus GST-p107 R1-R2 protein, amounts of the purified proteins used should be provided and not expressed in microliters (Figure 4A). More importantly, in co-transfection assays in 293 cells using a similar Flag tag for p107 and FAM122A, it seems that at identical expression levels (Figure 4C, left panel, lane 50), B55a do not preferentially interact with FAM122A, arguing against the proposed model by the authors.

. Experiments on cell cycle progression defects upon FAM122A inactivation are too preliminary and do not allow to conclude on a delay in S phase onset, in S progression or both. Indeed, a delay in AKT and GSK3b phosphorylation and CyclinD1 expression is observed in serum starvation/ re-stimulation experiment upon FAM122A inactivation (Figure 7D), which might suggest that cells are delayed in G1 phase. Conversely, the timing of Cyclin E1 expression, which is required to trigger S-phase onset, is not perturbed (Figure 7B). To accurately monitor progression from G1 to S phase upon serum stimulation, EdU or BrDU labelling should be performed and is crucially missing in Figure 7A.

. Finally, it is unclear if FAM122A inactivation abrogates S-phase checkpoint upon thymidine or HU treatment, as proposed by the authors. Indeed, if the cells do not activate DNA replication checkpoint and failed to arrest, Cyclin A, whose expression increases until mitosis (Pagano et al. EMBO 1992), should accumulate in FAM122A cells, which is not observed (Figure 8B).

One point that was not explored is at which cell cycle stage FAM122A KO cells die (subG1, Figure 8C) under HU treatment, which might help to better understand the contribution of FAM122A upon replication stress.

Other:

Raw data in Figures 7C, 8D and supp 7A should be provided.

RESPONSE TO REVIEWERS

Reviewer #1 (Remarks to the Author):

Although FAM122A has been proposed as a PP2A/B55 α holoenzyme inhibitor previously, there is really a lack of the evidence for the structural analysis of FAM122A interacting or inhibiting PP2A/B55 α associated with its substrates. In this study, the authors studied the main interaction regions of FAM122A with PP2A/B55 α by computational structure prediction and mutational and biochemical methods, and showed that conserved SLiM is necessary for FAM122A binding to B55 α in vitro and in cells, and also showed the spatial structure prediction of how FAM122A interrupts the interaction of B55 α and its substrate P107. These data are novel for this field and essential for the understanding the biology of FAM122A. Overall, the study is well-designed and supported by most experimental data.

We agree and thank the reviewer for their positive views on the novelty and significance of our work to the field, as well as the reviewer's comments that have helped us improve the manuscript.

Major comments:

1. "Whether the SLiM is critical for the inhibitory effect of FAM122A KO on cell growth is not very convince. In Fig. 6, the rescue of the growth inhibitory effect in FAM122A KO should be compared in the FAM122A KO cells re-expressed by FAM122A WT and FAM122A SLiM mutant cells."

We thank the reviewer for raising this concern. We noticed a color palette swap between the Parental and KO that may have contributed to some confusion from previous panels. We separated the WT and MUT FAM122A reconstitutions while showing the same Parental and KO data for comparison. We have also added shaded lines highlighting the difference between the parental and KO cells. We hope this addresses this concern.

2. Also, the reviewer wondered "if there is any try for the crystal structure experiments of FAM122A interacting with PP2A/B55 α with or without substrate."

While this would be an exciting advancement to our study, we find this to be beyond the scope. Additionally, another group has been pursuing this avenue based on our unpublished observations and FAM122A DNA construct and published a pre-print this September (see Padi et al. 2023 Preprint in Biorxiv- <https://doi.org/10.1101/2023.08.31.555365>). It should be noted that our AlphaFold-Multimer v2.3 model superposes on their Cryo-EM structure (PDB: 8S00) with high accuracy (1.00 Å RMSD over 1231 residues). It was shared with them and the research community in March 2023, well ahead of the Cryo-EM data deposition in the PDB (indeed the cryo-EM coordinates were only made publicly available less than one month ago, on October 25, 2023).

3 and 4: “In Fig.9D, FAM122A signaling is too weak and invisible. Is there any difference for DNA damage response between FAM122A KO and mGFP FAM122A upon HU treatment? Here just show one field and not clear the total effects.” and “Is the SLiM is necessary for the protection of FAM122A-expressing cells challenged by HU treatment. Or giving the discussion for any other possibility?”

We thank the reviewer for bringing this to our attention and for posing the question concerning the SLiM-dependent effects of HU treatment of the cells. We have increased the magnification and brightness of the presented images and have performed quantitative γ -H2AX intensity immunofluorescence experiments using the WT/SLiM Mutant reconstituted cells (new figure 9E with representative images in suppl Fig 9C). We note significant differences in the overall γ -H2AX intensity when the SLiM is WT vs mutant.

5. “There are several missing data in manuscript. Such as missing Suppl. Fig.3D, no Suppl. Fig. 4?”

We thank the reviewer for bringing this to our attention. Some of the Figures in the original manuscript did not have supplementary data. In addressing the Reviewer’s concerns, we have now rewritten the manuscript to include supplemental figures to all the main figures.

Some minor points:

1. “What’s the meaning of the symbols, such as “.”, “:” and “*” in Fig. 1.”

In Figure 1D, the symbols are used by UniProt to illustrate similarity within sequence alignments. We have added this information to the figure legend and below.

An “*” (asterisk) indicates positions that have a single, fully conserved residue.

A “:” (colon) indicates conservation between groups of strongly similar properties - scoring > 0.5 in the Gonnet PAM 250 matrix.

A “.” (period) indicates conservation between groups of weakly similar properties - scoring =< 0.5 in the Gonnet PAM 250 matrix.

2. “In Fig.3, the SLiM “RLHQIK” is not the same as stated in manuscript”.

The FAM122A RLHQIK SLiM shown in Figure 3, is a conserved version of the [RK]-[VIL]-x-x-[VIL]-[RK] degenerate consensus based on the [RK]-[V-x-x-[VI]-R SLiM that we described in Fowle et al. 2021 (eLife: 10.7554/eLife.63181).

3. “In Fig.7D, the annotations of pGSK and pERK in the right of the panels should be properly indicated.”

Thank you for bringing this to our attention. We have corrected the issue.

4. In all western blot, the molecular weights should be marked.

As the western blots show strips of the membranes, some of the molecular weight markers reside outside of the areas shown. We will mark the molecular weight of the markers in the supplementary full-length westerns that are required in the final version of the manuscript.

Reviewer #2 (Remarks to the Author):

“The manuscript by Wasserman et al, “Fam122A ensures cell cycle interphase progression and checkpoint control as a SLiM-dependent substrate-competitive inhibitor to the B55alpha/PP2A phosphatase, presents a lot of interesting data and modeling concerning the relationship of FAM122A to B55alpha/PP2A. The first part of the paper investigates how FAM122A binds to the B55alpha subunit and convincingly demonstrates that this interaction is mediated by a conserved ‘SLiM’, which actually seems to form an alpha helix (so is it actually a SLiM or more broadly defined as a docking motif?). While there is no structure presented, there is extensive modeling presented with different versions of alpha-fold and extensive mutagenesis and interaction assays (in vivo, in vitro, competition with p107 which contains the same docking motif) etc. This is the strongest part of the paper, and demonstration of key residues in B55alpha that are required for the interaction w/ FAM122A together with mutants of FAM122A make a very strong case for this ‘SLiM’-dependent interaction. The second part of the model proposes that a longer helix C-terminal to the B55alpha docking helix forms a direct interaction with the active site of the PP2A catalytic subunit. This part of the model is much less convincing as two mutants (conserved E residues changed to K) have no effect on observed FAM122A-B55alpha/PP2A interactions (through qualitative pull downs) and were not tested for effects on in vitro dephosphorylation. However, this paper successfully challenges an earlier study that proposed FAM122A inhibited B55alpha/PP2A by promoting degradation of the catalytic subunit.

Finally, the last part of the paper analyses in vivo effects of FAM122A deletion/re-expression with SLiM mutants. Effects on proliferation are clear (although were demonstrated in the 2016 paper). More interesting are demonstrated effects of FAM122A on cell cycle entry (via regulation of p107/RB protein phosphorylation) and regulation of G1/S checkpoint control during replicative stress. However, localization analyses proposed to demonstrate that FAM122A localizes exclusively to the nucleus during interphase, are not (in current form) convincing.

Overall, the paper has important findings but the overall impact is reduced by significant flaws. The paper is quite long and could benefit from eliminating some redundancies to focus on the key experiments that provide the most informative and novel results and indication of mechanisms for observed effects in vivo. See below for specific comments and suggestions for modifications to strengthen the study.”

We thank the reviewer for their assessment of our findings on the mechanism of PP2A/B55 α inhibition by FAM122A and thank the reviewer for comments on how to improve the manuscript with respect to the inhibitory helix and the biological data.

1. “In Figure 1 authors present their pipeline for identifying proteins containing the proposed B55alpha binding ‘SLiM’ which combines bioinformatic analyses with cross validation with several published experimental data sets. While it is clear that FAM122A is the ‘winner’ from these analyses some discussion of other candidates that came out from these extensive

analyses should also be included. In particular, candidates that in figure 1B were identified in 4/5 of 5/5 data sets should be listed along with their proposed SLiMs and identifying which were previously identified (i.e. p107, etc.). Currently supplemental table 1 only contains information about phosphoproteomic analyses carried out by these authors.”

We thank the reviewer for the recommendation. We have included a new Supplemental Table 1 that provides the common proteins in each dataset with a potential SLiM and its sequence.

2. “Mutagenesis of the ‘SLiM’ and complementary mutagenesis of B55 alpha supports the predicted model strongly. Authors state, however that S73A mutation abolishing a phosphosite amino-terminal to the SLiM had no effect on interaction. Confidence in this statement is reduced by the fact that no quantitative assays are performed to determine affinity of the wt FAM122A or mutants for B55alpha/PP2A, and there is no evidence presented or discussed concerning whether this site is ever phosphorylated, and if the wild-type FAM122A would have been phosphorylated at this site under co-IP conditions used to test effects of the S73A mutant. In general, the lack of quantitative affinity determinations are a weakness in the paper.”

We thank the reviewer for their critique and have provided quantitation for the effects of phosphomimetics (murine S73E) and non-phosphorylatable (murine S73A) on B55alpha binding compared to WT and SLiM mutants for immunoprecipitation experiments (please see the new Suppl. Figure 2A). Very minor changes, that are not significant are observed. Additionally, we note that the human site, S76, is detected to be significantly dephosphorylated in the presence of B55alpha overexpression (Fig. 1C), which may allow dephosphorylation by free PP2A/B55 α in trans. This and other Ser-Pro directed sites can be phosphorylated *in vitro* by Cyclin A/CDK2, Cyclin B/CDK1, and ERK1 (our data not shown). However, consistently with FAM122A inhibitory activity on PP2A/B55 α , none of the detected sites are effectively dephosphorylated by PP2A/B55 α in the presence of excess FAM122A.

3. “In Figure 2 analysis of conserved regions is presented. FR4 contains BOTH predicted helices as shown in the next section of the paper and this should be emphasized. There is no direct evidence that the second helix forms or that it interacts w/ catalytic active site of PP2AC, which is the obvious missing link in the authors’ proposed mechanism. Besides modeling, there is no convincing support presented. Analyses of FAM122A binding to B55alpha WT vs mutant C239G are consistent with FAM122A binding to the holoenzyme through additional interactions, but this is an indirect argument and does not identify where the additional interactions may be. Furthermore a potentially interesting effect on FAM122A phosphorylation is pointed out (Figure 3E), but not explained or discussed further. Rather, direct experiments that examine any effects of mutations proposed to disrupt this second helix on FAM122A-B55alpha/PP2A binding and/or dephosphorylation should be included. Most important, authors propose that E97 and E101 contact Mn²⁺ residues in the catalytic active site and that changing these to K does not affect qualitative assays of FAM122A-B55alpha PP2A binding. To identify any ability of FAM122A to inhibit the catalytic center of PP2AC, authors should examine *in vitro* dephosphorylation of a small molecule (pNPP or

DiFMUP) that does not depend on SLiM-dependent docking, by the B55alpha heterotrimer +/- FAM122A wt or E97,101K mutants.”

We thank the reviewer for their critique and great suggestions! We have rewritten the section to emphasize that FR4 contains BOTH helices. We have also taken the reviewer's suggestions and performed mutagenesis of murine FAM122A E97 and E101 (which correspond to E100 and E104 in humans) to either Proline or Alanine. These Proline residues should disrupt the helix, whereas Alanine would maintain the structure. Furthermore, we used these mutants in DiFMUP phosphatase assays and compared their inhibitory effect to WT (please see new Figure 4E). Overall, we observe statistically significant impairment of PP2A/B55a inhibition in the presence of these mutants. Again, we thank the reviewer for this most insightful suggestion.

4. “Discussion of different versions of AlphaFold structural predictions make up a large portion of the paper. Since the newest version of AlphaFold Multimer v2.3 seems to provide the most comprehensive information, I suggest editing the paper to contain a single section where the most confident structural prediction is presented and discussed.”

We have taken the reviewer's suggestion and have restructured the presentation of the AlphaFold2 Multimer data and explanations used in the manuscript, which has resulted in rearrangement of Figs. 3-5).

5. “The next part of the paper analyses in vivo effects of FAM122A on proliferation/colony formation (Fig6), and shows it is SLiM dependent. This strongly supports the structural model and demonstration that p107 and FAM122A compete for binding to B55alpha/PP2A holoenzyme.”

We thank the reviewer for the praise and agree with the assessment.

6. In Figs. 7, 8 and 9, authors investigate cell cycle entry/progression, as well as effects on G1/S checkpoint activation in cells lacking FAM112A. These are complicated in vivo assays and would be easier for a non-expert to appreciate by highlighting what the direct B55alpha/PP2A substrates are in each case, and if deletion of FAM112A causes the observed biological effects through increased activity of the phosphatase as one would predict from deletion of an inhibitor. As is, the studies seem mainly descriptive.

We thank the reviewer for bringing this to our attention. We have highlighted potential substrates in the text. Concerning an increase in activity specific to B55, we attempted to perform DiFMUP phosphatase assays from the immunoprecipitations of Parental and KO cells using various anti-B55 antibodies and failed to detect differences, which may be a limitation of the antibodies used. We assessed the total phosphatase activity in Suppl. Fig 6 using phosphospecific antibodies directed towards SP and TP-directed CDK substrates and observed faster dephosphorylation kinetics. Also, The SLiM dependent reconstitution assays support the idea that the proliferation and checkpoint defects are SLiM dependent.

7. "Figure 9 presents potentially interesting analyses of FAM122A localization, and identifies an NLS. However, in Figure 9A images, it is very difficult to see RFP-FAM122A signal and similarly in figure 9D, FAM122A and gammaH2Ax signals are barely detectable and would be better appreciated perhaps if shown as high contrast black/white images. Finally, no information is presented about any phenotypes observed when the NLS is mutated, so it is difficult for the reader to appreciate the potential biological relevance of the reported nuclear localization of FAM122A."

We thank the reviewer for their comments. We have changed Fig 9A to show a cell with higher visibility of RFP-FAM122A and have split the channels to represent the changes in localization better. We have also reviewed and magnified the image from Fig 9D and believe the images are now brighter and of higher quality. We generated the NLS-mutants to further support the importance of the localization data. FAM122A is nuclear through interphase. And in response to HU treatment colocalizes with γ H2AX (improved Fig. 9D). We have also shown that the DNA damage induced by HU in 293-FAM122A KO cells is reduced by reconstitution of WT, but not the SLiM mutant (Fig. 9E). We agree with the reviewer that the identification of the NLS is interesting, but we did not establish the necessary stable cell lines to address phenotypes. Future experiments beyond the scope of this manuscript could determine if FAM122A participates in B55 α recruitment to the nucleus and whether FAM122A controls any PP2A/B55 α substrates outside of the nucleus, which could potentially happen at some point in mitosis.

Reviewer #3 (Remarks to the Author):

"In the manuscript entitled "FAM122A ensures cell cycle interphase progression and checkpoint control as a SLiM-dependent substrate-competitive inhibitor to the B55 α /PP2A phosphatase", the authors explore the biological functions of FAM122A protein, a proposed inhibitor of the phosphatase PP2A/B55a holoenzyme. The manuscript is separated in two parts, the first one being mostly dedicated to identify amino-acid residues involved in FM122A - PP2A/B55a interaction and the second one investigating cell cycle defects upon FAM122A knock-out.

Overall, I found this work interesting, but I have several concerns about the main conclusions, as detailed below.

Also, I found this manuscript difficult to read as figure legends frequently do not describe/explain the different conditions used."

We thank the reviewer for the positive views on the manuscript and the comments that have allowed us to improve the manuscript.

Main points:

“The authors propose that FAM122A acts as a competitive inhibitor and exhibits a higher affinity for PP2A/B55a than other substrates, such as RB-related protein p107.

First, it is unclear in Figure 2E if the authors compared (by GST pulldown) the affinity for B55a of a full-length FAM122A (FL) protein versus a truncated p107 construct (p107 R1-R2). Second, to conclude on in vitro competition assays using purified HA-FAM122A versus GST-p107 R1-R2 protein, amounts of the purified proteins used should be provided and not expressed in microliters (Figure 4A). More importantly, in co-transfection assays in 293 cells using a similar Flag tag for p107 and FAM122A, it seems that at identical expression levels (Figure 4C, left panel, lane 50), B55a do not preferentially interacts with FAM122A, arguing against the proposed model by the authors.”

We thank the reviewer for their comments. We have measured the affinity of FAM122A and p107 R1R2 by depletion and found FAM122A to bind with a 2.4-fold higher affinity than p107 R1R2 in the conditions used (please see Suppl Fig 5). p107 is the best substrate binder that we have tested over the years. Under the same experimental conditions, it binds relatively better than pRB, much better than KSR, and much better than other reported PP2A/B55 α substrates that do not bind under our experimental conditions. We have also corrected the *in vitro* competition experiments to show the amount of protein used rather than volume. With regard to the in-cell competition assays, the assay has some limitations, as there might not be enough expressed p107 to bind all available B55 α in the cell. In that case, a portion of the increasing amount of FAM122A could bind B55 α that is not accessible to p107. Thus, the fact that FAM122A displaces p107 at concentrations that are comparable suggests tighter binding.

“Experiments on cell cycle progression defects upon FAM122A inactivation are too preliminary and do not allow to conclude on a delay in S phase onset, in S progression or both. Indeed, a delay in AKT and GSK3b phosphorylation and CyclinD1 expression is observed in serum starvation/ re-stimulation experiment upon FAM122A inactivation (Figure 7D), which might suggest that cells are delayed in G1 phase. Conversely, the timing of Cyclin E1 expression, which is required to trigger S-phase onset, is not perturbed (Figure 7B). To accurately monitor progression from G1 to S phase upon serum stimulation, EdU or BrDU labelling should be performed and is crucially missing in Figure 7A.”

We thank the reviewer for their comments and suggestions. We have addressed the comments by performing the EdU incorporation of T98G cells (Parental and KO) in the new Fig 7C. Here, we observe slower progression from G1 to S in the FAM122A KO cells, which is consistent with the delays in the phosphorylation of early mitogenic signaling kinases (AKT1 and ERK1), cyclin D1 expression, pRB phosphorylation and the expression of E2F-regulated gene products (p107 and cyclin A). We agree with the reviewer that changes in cyclin E are not clear and have removed that statement in the text. We have also performed DNA fiber assays using HEK293 cells. We found the overall fiber length of CldU and IdU incorporation to be less in the FAM122A KO cells than in the parental (see new Fig 8B), demonstrating delays in DNA replication. Therefore, we detect defects in proliferation that are associated with G1 progression in T98G cells, which have a functional restriction point and in S phase in HEK293 cells, which do not, because the cells express E1A inactivating pRB proteins.

“Finally, it is unclear if FAM122A inactivation abrogates S-phase checkpoint upon thymidine or HU treatment, as proposed by the authors. Indeed, if the cells do not activate DNA replication checkpoint and failed to arrest, Cyclin A, whose expression increases until mitosis (Pagano et al. EMBO 1992), should accumulate in FAM122A cells, which is not observed (Figure 8B).

One point that was not explored is at which cell cycle stage FAM122A KO cells die (subG1, Figure 8C) under HU treatment, which might help to better understand the contribution of FAM122A upon replication stress.”

We thank the reviewer for these comments that will allow us to clarify the data. Thymidine challenge experiments have been repeated multiple times and we do not observe accumulation of cell at the G1/S transition. Consistently cell cycle markers remain unchanged, indicating lack of accumulation of cells at any cell cycle phase (Fig. 8C), which is observed in the WT cells. In addition, this defect is rescued in by FAM122A in a SLiM-dependent manner (Fig. 8E). HU causes activation of CHK2 and CHK1, consistent with accumulation of cyclin E, which is not observed in the FAM122A KO cells, which show attenuated checkpoint signaling (lower CHK2 and CHK1 phosphorylation), and results in DNA damage. Furthermore, given that the effects on accumulation of γ H2AX are detected just 3-6 hours following treatment, it appears that cell are dying in S phase.

“Other:

Raw data in Figures 7C, 8D and supp 7A should be provided.”

These data are provided as requested.

Reviewers' Comments:

Reviewer #1:

Remarks to the Author:

The author has responded to all my comments. I have no further questions. I think the revised manuscript is improved a lot compared with the previous one and suitable for the publication.

Reviewer #3:

Remarks to the Author:

Some additional data have been provided in the revised version that reinforce to some extent the original manuscript by Wasserman et al.

Nonetheless, the experiments on cell growth perturbations upon FAM122A KO appear a bit dispersed. Notably, experiments combining FAM122A KO and replication stress upon HU or thymidine treatment are preliminary and difficult to interpretate. Importantly, it should be indicated in the figure legends how many experiments have been performed each time (N= ?). Altogether, I agree with the comments from reviewer 2 that the manuscript is quite long and that in the second part on cell growth perturbations it will benefit to be more focused on the key experiments that provide robust information on potential deregulations induced upon FAM122A depletion, as discussed below.

The data provided show that the depletion of FAM122A globally slow down cell proliferation (Figure 6). Consistently, investigating these effects in more details,

(i) the authors found that the cell cycle resumption is slow-down by around 4 hours in FAM122A depleted cells following a serum starvation – release regime.

(ii) that incorporation of BrdU is reduced during S-phase in exponentially growing cells in absence of FAM122A. Supporting this observation the authors provided a new result showing that replication fork progression by DNA combing and CldU – IdU labelling is slow-down.

Because FAM122A is proposed to act as an inhibitor of B55a/PP2A phosphatase that counteracts Cdk-dependent phosphorylation of protein targets, one possible scenario is that overall Cdk activity is lower during S-phase upon FAM122A depletion affecting replication origin firing and/or fork progression. This scenario has not been investigated in the manuscript. It will have been of interest to analyze in the DNA combing experiment provided (N=1 experiment ?) the density of replication forks. Could it be possible to restore BrdU incorporation in S phase cells upon FAM122A depletion by partial Wee1 inhibition (MK-1775) ?

Concerning the experiments combining FAM122A depletion with replication stress upon HU or excess of thymidine, I found, as already mentioned, that their interpretation is difficult and require more investigation. The authors report that FAM122A KO cells fail to arrest (in S phase) upon thymidine treatment. Conversely, is there a cell cycle arrest in the next G2 or G1 phase ?, that might explain the persistence of G1 and G2 populations by flow cytometry under this treatment (Figure 8C). Did the authors explore by Western blot the expression of p53 and p21 proteins in these conditions? Indeed, if FAM122A depletion already affects S-phase progression promoting a mild replication stress, additional replication stress upon HU or Thymidine treatment might strongly promote p53-dependent cell cycle withdrawal explaining the distribution profile by FACS and the appearance of a pronounced subG1 population as shown on Figure 8D.

Conversely, the reported effects of FAM122A depletion on checkpoint activation upon replication stress are puzzling. For instance, Chk1 phosphorylation on Ser345 following HU treatment is reduced in FAM122A depleted cells (Figure 8D) but conversely the migration shift due to Chk1 phosphorylation is similar upon or not FAM122A depletion on the same figure (Figure 8D).

Of interest, a lower activation of checkpoint signaling upon FAM122A depletion might be a direct consequence of a lower number of active replication forks in these cells that remain to investigate.

Reviewer #4:

Remarks to the Author:

The authors did a commendable job addressing reviewer #2 critiques.

However, one significant issue remains. Specifically, the group states in the Methods section that data were analyzed by either T-test or ANOVA, but for the most part does not specify in the figure legends which test(s) were carried out where.

The one exception is the legend to Fig. 9, where the authors state that a t-test was used to analyze the experiment in Fig. 9E. This bar graph lists three conditions with two comparisons so has to be analyzed by ANOVA (with Dunnett's posthoc test). A t-test is inappropriate here.

Throughout the manuscript, the authors need to explicitly state the tests applied, considering the number of conditions present and also whether data fulfill the requirements for parametric tests.

WASSERMAN et al RESPONSE TO REVIEWERS' COMMENTS

The reviewer comments are within quotation marks and our responses are bolded.

Reviewer #1 (Remarks to the Author):

“The author has responded to all my comments. I have no further questions. I think the revised manuscript is improved a lot compared with the previous one and suitable for the publication.”

We thank the reviewer for their suggestions and for facilitating the improvement in the quality of the manuscript.

Reviewer #4 (Remarks to the Author):

“The authors did a commendable job addressing reviewer #2 critiques.”

We agree, we thank the reviewer for acknowledging our efforts. We also thank reviewer #2 for bringing up concerns that have helped us improve manuscript.

“However, one significant issue remains. Specifically, the group states in the Methods section that data were analyzed by either T-test or ANOVA, but for the most part does not specify in the figure legends which test(s) were carried out where.”

Thanks for bringing this to our attention. This was an oversight and has been added to the figure legends.

“The one exception is the legend to Fig. 9, where the authors state that a t-test was used to analyze the experiment in Fig. 9E. This bar graph lists three conditions with two comparisons so has to be analyzed by ANOVA (with Dunnett's posthoc test). A t-test is inappropriate here.”

We apologize, there was a misprint as these data were analyzed using Brown-Forsythe and Welch ANOVA test using Dunnett's T3 multiple comparisons test. We have since looked at the normality of the data and subsequently used a using a Kruskal-Wallis test with a Dunn's multiple comparison test.

“Throughout the manuscript, the authors need to explicitly state the tests applied, considering the number of conditions present and also whether data fulfills the requirements for parametric tests.”

Thank you. We have looked at the normality of the data and determined the appropriate statistical tests, which has been indicated in the figure legends.

Reviewer #3 (Remarks to the Author): **Reviewer #3 still had some concerns and questions that we address below:**

“Some additional data have been provided in the revised version that reinforce to some extent the original manuscript by Wasserman et al.

Nonetheless, the experiments on cell growth perturbations upon FAM122A KO appear a bit dispersed. Notably, experiments combining FAM122A KO and replication stress upon HU or thymidine treatment are preliminary and difficult to interpretate.

Importantly, it should be indicated in the figure legends how many experiments have been performed each time (N= ?).”

Thank you. The number of replicates was referenced in the *Graphs and Statistical Analysis* section of the methods and stated experiments were performed in triplicate unless otherwise stated. We have now made sure that *N* is stated in the figure legends, when relevant.

“Altogether, I agree with the comments from reviewer 2 that the manuscript is quite long and that in the second part on cell growth perturbations it will benefit to be more focused on the key experiments that provide robust information on potential deregulations induced upon FAM122A depletion, as discussed below.”

The length of the manuscript was addressed per reviewer 2’s initial comments. To accomplish that we restructured the description of the data leading to the final mechanism of inhibition, including a reorganization of the corresponding Figs. and made other changes through the text.

“The data provided show that the depletion of FAM122A globally slow down cell proliferation (Figure 6). Consistently, investigating these effects in more details,
(i) the authors found that the cell cycle resumption is slow-down by around 4 hours in FAM122A depleted cells following a serum starvation – release regime.
(ii) that incorporation of BrdU is reduced during S-phase in exponentially growing cells in absence of FAM122A. Supporting this observation the authors provided a new result showing that replication fork progression by DNA combing and CldU – IdU labelling is slow-down.”

Because FAM122A is proposed to act as an inhibitor of B55a/PP2A phosphatase that counteracts Cdk-dependent phosphorylation of protein targets, one possible scenario is that overall Cdk activity is lower during S-phase upon FAM122A depletion affecting replication origin firing and/or fork progression. This scenario has not been investigated in the manuscript.

Thank you for this comment. The main focus of our manuscript is to describe the mechanism by which FAM122A inhibits PP2A/B55 α via direct competition with substrates and linked occlusion of the active site and the resulting effects in cell cycle progression in G1 and S and checkpoint activation to demonstrate the biological significance of our findings concerning the dependency on the helical motif. The effects of upregulating PP2A/B55 α as result of FAM122A depletion are pleotropic because it likely affect multiple substrates. As the reviewer suggests, one possibility is reduced CDK activity. Strongly supporting this, we point to our new data showing diminished cyclin E expression in both untreated and thymidine/HU treated FAM122A KO cells compared to parental HEK293 cells (Fig. 8C (new) and 8D). These data strongly indicate that the activity of the CDK2/CDK1 associated with cyclin E should be dramatically diminished.

It will have been of interest to analyze in the DNA combing experiment provided (N=1 experiment ?) the density of replication forks.

Thanks for this comment as it helps us clarify the approach. The data for the DNA fiber assay shown in Fig. 8B consists of three independent replicates, where all the measurements are pooled together. We have clarified this in the figure legend. These data demonstrate reduced rates of DNA synthesis. To determine the density of replication forks from these assays would require a prerequisite analysis of the number of cells in S-phase with sufficient cell counts to mitigate any inequalities and cannot adequately be assessed from the previously performed

experiment. We believe that this analysis, although interesting, is beyond the scope of the manuscript.

Could it be possible to restore BrdU incorporation in S phase cells upon FAM122A depletion by partial Wee1 inhibition (MK-1775)?

Thanks for the question. Such results have been shown in Li et al. 2020 Supplemental Figure 3D in A549 cells using AZD-1775 (MK-1775).

Concerning the experiments combining FAM122A depletion with replication stress upon HU or excess of thymidine, I found, as already mentioned, that their interpretation is difficult and requires more investigation. The authors report that FAM122A KO cells fail to arrest (in S phase) upon thymidine treatment. Conversely, is there a cell cycle arrest in the next G2 or G1 phase?, that might explain the persistence of G1 and G2 populations by flow cytometry under this treatment (Figure 8C).

We thank the reviewer for these comments. We believe that the new experiments added to solve this question bring clarity to the effects of FAM122A KO in checkpoint activation. We have performed new thymidine challenge experiments adding nocodazole to replicate samples to determine the proportion of cells progressing through mitosis in the FAM122A KO cells (Fig. 8C is completely new). We have also extended time points to ensure enough time for cells to move clearly to exit mitosis into the next cycle.

The new Figure 8C shows a thymidine challenge that resulted in time-dependent accumulation of HEK293 cells at the G1/S transition and in S phase that correlated with accumulation of Cyclin E peaking at 18 h. By contrast, FAM122A-KO cells exhibited diminished sensitivity to nucleotide depletion and although they slowed down through S phase, they continued to progress through G2 and completed mitosis. Completion of mitosis in FAM122A KO cells was demonstrated by adding nocodazole to replicate samples 12 hours post washout, which resulted in massive accumulation of cells in G2/M with very few cells remaining in S phase by 24 h (Fig 8C, note gray overlay over the untreated KO cells). This was followed by massive cell death (sub-G1 DNA-accumulation at 32 hrs). Progression of KO cells through S/G2/M was also clear from the accumulation of cyclins A and B that peaked at 18 and 24 hrs respectively.

“Did the authors explore by Western blot the expression of p53 and p21 proteins in these conditions? Indeed, if FAM122A depletion already affects S-phase progression promoting a mild replication stress, additional replication stress upon HU or Thymidine treatment might strongly promote p53-dependent cell cycle withdrawal explaining the distribution profile by FACS and the appearance of a pronounced subG1 population as shown on Figure 8D.

We thank the reviewer for these comments. We did not previously explore p53-related mechanisms and its target p21 by western blot under the conditions of nucleotide deprivation as HEK293 cells were originally established by immortalization with adenoviral genomes and express E1A and E1B. E1B has been shown to inhibit p53 (Debbas and White, 1993, Martin and Berk, 1998, Querido et al. 2001) and therefore HEK293 cells are believed to have an inhibited p53 pathway.

To our surprise we have detected time-dependent accumulation of p53 in FAM122A KO cells, which started as cells were moving to G2/M and was followed by accumulation of p21 in G1 (this shows that p53 is not completely inactive in HEK293 cells). p21 levels increase at 24 and 32 h, but this increase is blocked by nocodazole, indicating that p21 expression is upregulated in

late mitosis or G1 (Fig. 8C). This accumulation of p53 and p21 may slow down the FAM122A KO cells in the next G1 phase. The p53 upregulation may also contribute to the cell death observed in the presence of nocodazole.

Conversely, the reported effects of FAM122A depletion on checkpoint activation upon replication stress are puzzling. For instance, Chk1 phosphorylation on Ser345 following HU treatment is reduced in FAM122A depleted cells (Figure 8D) but conversely the migration shift due to Chk1 phosphorylation is similar upon or not FAM122A depletion on the same figure (Figure 8D).

The migration shifts shown are likely the result of differential phosphorylation on the Chk1 protein. Chk1 Ser345 is a single site that is referenced for activation. Other phosphorylation sites such as Ser317 or its autophosphorylation at Ser296 may be responsible for differences in migration species.

Of interest, a lower activation of checkpoint signaling upon FAM122A depletion might be a direct consequence of a lower number of active replication forks in these cells that remain to investigate.

We agree with the reviewer's sentiment as it would warrant a follow-up but it is beyond the scope of the submitted manuscript.

Reviewers' Comments:

Reviewer #4:

Remarks to the Author:

My relatively minor concerns about the statistical analyses have been addressed in the revised MS.